# AQuaMaM: An Autoregressive, Quaternion Manifold Model for Rapidly Estimating Complex $\mathbf{SO}(3)$ Distributions

## Abstract

Accurately modeling complex, multimodal distributions is necessary for optimal decision-making, but doing so for rotations in three-dimensions, i.e., the $\mathbf{SO}(3)$ group, is challenging due to the curvature of the rotation manifold. The recently described implicit-PDF (IPDF) is a simple, elegant, and effective approach for learning arbitrary distributions on $\mathbf{SO}(3)$ up to a given precision. However, inference with IPDF requires $N$ forward passes through the network's final multi-layer perceptron—where $N$ places an upper bound on the likelihood that can be calculated by the model—which is prohibitively slow for those without the computational resources necessary to parallelize the queries. In this paper, I introduce AQuaMaM,[1][2] a neural network capable of both learning complex distributions on the rotation manifold *and* calculating exact likelihoods for query rotations in a single forward pass. Specifically, AQuaMaM *autoregressively* models the projected components of unit quaternions as mixtures of uniform distributions that partition their geometrically-restricted domain of values. On an "infinite" toy dataset with ambiguous viewpoints, AQuaMaM rapidly converges to a sampling distribution closely matching the true data distribution. In contrast, the sampling distribution for IPDF dramatically diverges from the true data distribution, despite IPDF approaching its theoretical minimum evaluation loss during training. On a constructed dataset of 500,000 renders of a die in different rotations, an AQuaMaM model trained from scratch reaches a log-likelihood 14% higher than an IPDF model using a pretrained ResNet-50. Further, compared to IPDF, AQuaMaM uses 24% fewer parameters, has a prediction throughput $52\times$ faster on a single GPU, and converges in a similar amount of time during training.

## 1 Introduction and Related Work

In many robotics applications, e.g., robotic weed control (Wu et al., 2020), the ability to accurately estimate the poses of objects is a prerequisite for successful deployment. However, compared to other automation tasks, which primarily involve either classification or regression in $\mathbb{R}^n$, pose estimation is particularly challenging because the 3D rotation group $\mathbf{SO}(3)$[3] lies on a curved manifold. As a result, standard probability distributions (e.g., the multivariate Gaussian) are not well-suited for modeling elements of the $\mathbf{SO}(3)$ set. Further, because the steps for interacting with an object in the "mean" pose *between* two possible poses (Figure 1) will often fail when applied to the object when it is in one of the *non-mean* poses, accounting for multimodality in the context of rotations is essential.

The recently described implicit-PDF (IPDF) (Murphy et al., 2021) is a simple,

---

[1]Pronounced "aqua ma'am".

[2]All code to generate the datasets, train and evaluate the models, and generate the figures can be found at: <anonymized for review>.

[3]$\mathbf{SO}(3)$ stands for "special orthogonal group in three dimensions", with the "special" referring to the fact that all rotation matrices have a determinant of one. See: `https://blogs.scientificamerican.com/roots-of-unity/a-few-of-my-favorite-spaces-so-3/` for a popular science introduction to $\mathbf{SO}(3)$.

elegant, and effective approach for modeling distributions on $\mathbf{SO}(3)$ that both respects the curvature of the rotation manifold and is inherently multimodal. The IPDF model $f$ is trained through negative sampling where, for each ground truth image/rotation pair $(\mathbf{X}, \boldsymbol{R}_1)$, a set of $N - 1$ negative rotations $\{\boldsymbol{R}_i\}_2^N$ are sampled and a score is assigned to each rotation matrix as $s_i = f(\boldsymbol{X}, \boldsymbol{R}_i)$. The final density $p(\boldsymbol{R}_1|\mathbf{X})$ is then approximated as $p(\boldsymbol{R}_1|\mathbf{X}) \approx \mathrm{softmax}(\boldsymbol{s})[1]/V$ where $\boldsymbol{s}$ is a vector containing the scores with $\boldsymbol{s}[1] = s_1$, and $V = \pi^2/N$, i.e., the volume of $\mathbf{SO}(3)$ split into $N$ pieces. While effective, the $N$ hyperparameter induces

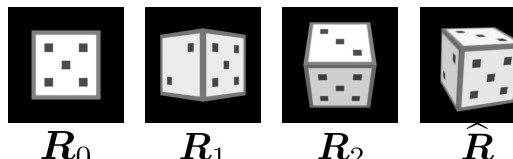

$\boldsymbol{R}_0 \qquad \boldsymbol{R}_1 \qquad \boldsymbol{R}_2 \qquad \widehat{\boldsymbol{R}}$

Figure 1: When minimizing the unimodal Bingham loss for the two rotations $\boldsymbol{R}_1$ and $\boldsymbol{R}_2$, the maximum likelihood estimate $\widehat{\boldsymbol{R}}$ is a rotation that was never observed in the dataset. Note, the die images are for demonstration purposes only, i.e., no images were used during optimization. $\boldsymbol{R}_0$ is the identity rotation.

a non-trivial trade-off between model precision (in terms of the maximum likelihood that can be assigned to a rotation) and inference speed in environments that do not have the computational resources necessary to extensively parallelize the task. Further, again due to the precision/speed trade-off, IPDF is trained with $N_{\mathrm{train}} \ll N_{\mathrm{test}}$,[4] which can make it difficult to reason about how the model will behave in the wild.

The alternative to *implicitly* modeling distributions on $\mathbf{SO}(3)$ is *explicitly* modeling them. Here, I briefly describe the three baselines used in Murphy et al. (2021), which are representative of explicit distribution modeling approaches. Notably, IPDF outperformed each of these models by a wide margin in a distribution modeling task (see Table 1 in Murphy et al. (2021)). First, Prokudin et al. (2018) described a biternion network (Beyer et al., 2015) trained to output Euler angles by optimizing a loss derived from the von Mises distribution. The two multimodal variants of the model consist of: (1) a variant that outputs mixture components for a von Mises mixture distribution, and (2) an "infinite mixture model" variant, which is implemented as a conditional (Sohn et al., 2015) variational autoencoder (Kingma & Welling, 2014). Second, Gilitschenski et al. (2020) described a procedure for training a network to directly model a distribution of rotations by optimizing a loss derived from the Bingham distribution (Bingham, 1974), with the multimodal variant outputting the mixture components for a Bingham mixture distribution. Lastly, Deng et al. (2020) extended the work of Gilitschenski et al. (2020) by optimizing a "winner takes all" loss (Guzmán-rivera et al., 2012; Rupprecht et al., 2017) in an attempt to overcome the difficulties (Makansi et al., 2019) commonly encountered when training Mixture Density Networks (Bishop, 1994).

One additional approach that needs to be introduced is the direct classification of an individual rotation from a set of rotations, which, in some sense, is the explicit version of IPDF. Unfortunately, the computational complexity of this strategy quickly becomes prohibitive as more precision is required. For example, Murphy et al. (2021) used an evaluation grid of $\sim$2.4 million equivolumetric cells (Yershova et al., 2010), which would not only require an extraordinary number of parameters in the final classification layer of a model, but would also require an extremely large dataset to reasonably "fill in" the grid due to the curse of dimensionality.[5]

In this paper, I introduce **AQuaMaM**, an **A**utoregressive, **Qua**ternion **Ma**nifold **M**odel that learns arbitrary distributions on $\mathbf{SO}(3)$ with a high level of precision. Specifically, AQuaMaM models the projected components of unit quaternions as mixtures of uniform distributions that partition their geometrically-restricted domain of values. Architecturally, AQuaMaM is a Transformer (Vaswani et al., 2017). Although Transformers were originally motivated by language tasks, their flexibility and expressivity have allowed them to be successfully applied to a wide range of data types, including: images (Parmar et al., 2018; Dosovitskiy et al., 2021), tabular data (Padhi et al., 2021; Fakoor et al., 2020; Alcorn & Nguyen, 2021c), multi-agent trajectories (Alcorn & Nguyen, 2021a), and proteins (Rao et al., 2021). Recently, multimodal Transformers (Ramesh et al., 2021; Reed et al., 2022), i.e., Transformers that *jointly* process different modalities of input (e.g., text and images), have revealed additional fascinating capabilities of this class of neural networks. AQuaMaM is multimodal in both senses of the word—it processes inputs from different modalities (images and unit quaternions) to model distributions on the rotation manifold with multiple modes.

---

[4]In Murphy et al. (2021), $N_{\mathrm{train}} = 4{,}096$ and $N_{\mathrm{test}} = 2{,}359{,}296$, i.e., $N_{\mathrm{train}}/N_{\mathrm{test}} = 0.2\%$.

[5]Notably, Mahendran et al. (2018) used a maximum of 200 rotations in their classification approach.

To summarize my contributions, I find that:

- AQuaMaM is highly effective at modeling arbitrary distributions on the rotation manifold. On an "infinite" toy dataset with ambiguous viewpoints, AQuaMaM rapidly converges to a sampling distribution closely matching the true data distribution (Section 4.1). In contrast, the sampling distribution for IPDF dramatically diverges from the true data distribution, despite IPDF approaching its theoretical minimum evaluation loss during training. Additionally, on a constructed dataset of 500,000 renders of a die in different rotations, an AQuaMaM model trained from scratch reaches a log-likelihood 14% higher than an IPDF model using a pretrained ResNet-50 (Section 4.2).

- AQuaMaM is *fast*, reaching a prediction throughput $52\times$ faster than IPDF on a single GPU.

## 2 THEORY

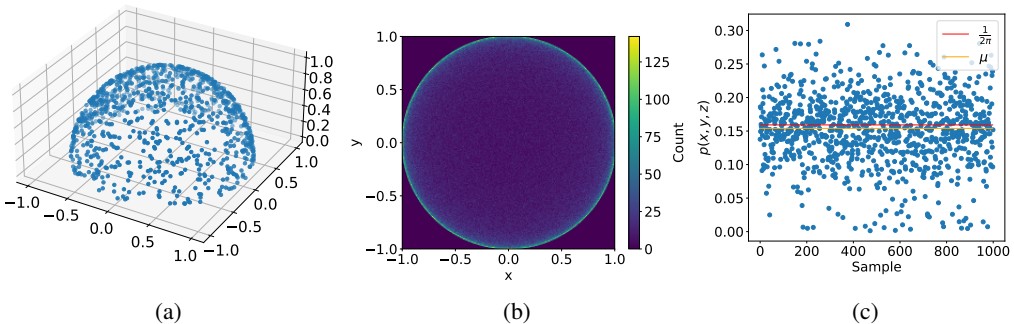

(a)                      (b)                      (c)

Figure 2: Because there is a bijective mapping between the unit disk $B^2 = \{v \in \mathbb{R}^2 : \|v\| < 1\}$ and the unit hemisphere $\widetilde{S}^2 = \{v \in \mathbb{R}^3 : \|v\| = 1, z > 0\}$, the challenging task of estimating a distribution on the curved $\widetilde{S}^2$ manifold can be simplified to estimating a distribution on the non-curved $B^2$. (a) Here, the true distribution on $\widetilde{S}^2$ is a uniform distribution, i.e., each point has a density of $\frac{1}{2\pi}$ (because $\widetilde{S}^2$ has a surface area of $2\pi$). (b) Points that are uniformly sampled from $\widetilde{S}^2$ and then projected onto $B^2$ are more concentrated towards the edges of $B^2$ due to the curvature of $\widetilde{S}^2$. If we model the distribution of $(x, y)$ coordinates on $B^2$ as a mixture of uniform distributions, we can calculate $p(x, y, z)$ by dividing $p(x, y)$ by the area of the parallelogram defined by the Jacobian located at $(x, y, z)$ on the hemisphere. (c) The $p(x, y, z)$ calculated through this procedure are generally quite close to the expected density. The mean density $\mu$ of the 1,000 points shown in (c) is 0.154 (compared to 0.159 for the true density). A similar procedure is used by AQuaMaM to obtain the probability of a unit quaternion $p(q)$ while only modeling the first three components of $q$: $q_x$, $q_y$, and $q_z$. See Section A.1 for additional details.

### 2.1 THE QUATERNION FORMALISM OF ROTATIONS

Several different formalisms exists for rotations in three dimensions, likely the most widely encountered of which is the rotation matrix—a $3 \times 3$ orthonormal matrix that has a determinant of one. Another formalism, particularly popular in computer graphics software, is the unit quaternion. Quaternions (typically denoted as the algebra $\mathbb{H}$) are extensions of the complex numbers, where the quaternion $q$ is defined as $q = q_x i + q_y j + q_z k + q_w$ with $q_x, q_y, q_z, q_w \in \mathbb{R}$ and $i^2 = j^2 = k^2 = ijk = -1$. Remarkably, the axis and angle of every 3D rotation can be encoded as a unit quaternion $q \in \mathbb{H}_1 = \{q \in \mathbb{H} : \|q\| = 1\}$, i.e., $q_w = \cos(\theta/2)$, $q_x = e_x \sin(\theta/2)$, $q_y = e_y \sin(\theta/2)$, and $q_z = e_z \sin(\theta/2)$ where $e = [e_x, e_y, e_z]$ is a unit vector indicating the axis of rotation and the angle $\theta$ describes the magnitude of the rotation about the axis.

### 2.2 MODELING A DISTRIBUTION ON A "HYPER-HEMISPHERE"

Due to the curvature of the 3-sphere $S^3 = \{v \in \mathbb{R}^4 : \|v\| = 1\}$, directly modeling distributions on $\mathbb{H}_1$ is difficult. However, two facts about $\mathbb{H}_1$ present an opportunity for tackling this challenging task. First, the unit quaternions $q$ and $-q$ encode the same rotation, i.e., $\mathbb{H}_1$ double covers $\mathbf{SO}(3)$.

As a result, we can narrow our focus to the subset of unit quaternions with $q_w > 0$, i.e., $\widetilde{\mathbb{H}}_1 = \{q \in \mathbb{H}_1 : q_w > 0\}$ (practically speaking, we can convert any $q$ in a dataset with $q_w < 0$ to $-q$). Second, because all $q \in \mathbb{H}_1$ have a unit norm, $q_x$, $q_y$, and $q_z$ must all be contained within the unit 3-ball $B^3 = \{v \in \mathbb{R}^3 : \|v\| < 1\}$, which is *not* curved. Therefore, $p(q_x, q_y, q_z)$ can be estimated using standard probability distributions. Importantly, $\forall q \in \widetilde{\mathbb{H}}_1$, $q_w$ is fully determined by $q_x$, $q_y$, and $q_z$ because of the unit norm constraint, i.e., there is a bijective mapping $f : B^3 \to \widetilde{\mathbb{H}}_1$ such that $f(q_x, q_y, q_z) = [q_x, q_y, q_z, q_w]$ where $q_w = \sqrt{1 - q_x^2 - q_y^2 - q_z^2}$. Together, these facts mean we can calculate $p(q)$ by simply applying the appropriate density transformation to $p(q_x, q_y, q_z)$.

Intuitively, thinking about the probability *density* of a point as its infinitesimal probability *mass* divided by its infinitesimal *volume*, the density $p(q_x, q_y, q_z)$ needs to be "diluted" by however much its volume expands when transported to $\widetilde{\mathbb{H}}_1$.[6] Specifically, the diluting factor is $1/s_q$ where $s_q$ is the magnitude of the wedge product of the columns of the Jacobian matrix $J$ for $f$. Similar to how the magnitude of the cross product of two vectors equals the area of the parallelogram defined by the vectors, the magnitude of the wedge product for three vectors is the volume of the parallelepiped defined by the vectors.[7] In this case, $s_q = 1/q_w$, i.e., the diluting factor is $q_w$.[8] Figure 2 visualizes an analogous density transformation for the unit disk and unit hemisphere.

### 2.3 Modeling SO(3) distributions as mixtures of uniform distributions on projected unit quaternions

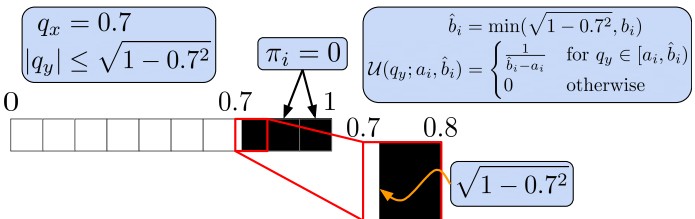

Figure 3: When modeling the conditional distribution $p(q_y|q_x)$ as a mixture of uniform distributions, the geometric constraints of the unit quaternion are easily enforced. Here, I focus on non-negative bins for clarity, i.e., intervals $[a_i, b_i]$ where $0 \le a < b \le 1$, but the same logic applies to negative bins. Given $q_x = 0.7$, we know that $|q_y| \le \sqrt{1 - 0.7^2}$ because $q$ has a unit norm. As a result, the mixture proportion $\pi_i$ for any bin where $\sqrt{1 - 0.7^2} < a_i$ *must* be zero. AQuaMaM enforces this constraint by assigning a value of $-\infty$ to the output scores for "strictly illegal bins" during training.[9] For the remaining bins, the corresponding uniform distribution is $\mathcal{U}(q_y; a_i, \hat{b}_i)$ where $\hat{b}_i = \min(\sqrt{1 - 0.7^2}, b_i)$, i.e., the upper bound of the uniform distribution for the partially legal bin is reduced to $\sqrt{1 - 0.7^2}$.

Using the chain rule of probability, we can factor the joint probability $p(q_x, q_y, q_z)$ as $p(q_x, q_y, q_z) = p(q_x)p(q_y|q_x)p(q_z|q_y, q_x)$. To model this distribution, desiderata for each factor include: (1) the flexibility to handle complex multimodal distributions and (2) the ability to satisfy the unit norm constraint of $q \in \widetilde{\mathbb{H}}_1$. By partitioning $[-1, 1]$ into $N$ bins, each with a width of $\frac{2}{N}$, we can model the distributions for $q_x$, $q_y$, and $q_z$ as mixtures of uniform distributions where $N$ controls the maximum precision of the model. Recall that the density for the uniform distribution $\mathcal{U}_{[a,b)}$ with $-\infty < a < b < \infty$ is $\mathcal{U}(x; a, b) = \begin{cases} \frac{1}{b-a} & \text{for } x \in [a, b) \\ 0 & \text{otherwise} \end{cases}$. Therefore, the density for $q_x$ is:

$$p(q_x) = \sum_{i=1}^{N} \pi_i \mathcal{U}(q_x; a_i, b_i) = \pi_k \mathcal{U}(q_x; a_k, b_k) = \pi_k \frac{1}{b_k - a_k} = \pi_k \frac{1}{2/N} = \pi_k \frac{N}{2} \tag{1}$$

---

[6]See the lecture notes here: https://www.cs.cornell.edu/courses/cs6630/2015fa/notes/pdf-transform.pdf for a survey of probability density transformations.

[7]See the lecture notes here: https://sheelganatra.com/spring2013_math113/notes/cross_products.pdf for additional connections between the cross product and the wedge product.

[8]See Section A.1 for the exact recipe.

[9]See Section A.2 for a subtlety regarding "strictly illegal bins".

where $\pi_i$ is the mixture proportion for the mixture component/bin indexed by $i$, and $k$ is the index for the bin that contains $q_x$.

For $q_y$ and $q_z$, the conditional densities are similar to the density for $q_x$, but the unit norm constraint on $\boldsymbol{q}$ presents opportunities for introducing inductive biases into a model (Figure 3). For example, consider the bins that are fully contained within $\mathbb{R}_{\geq 0}$, i.e., where $a, b \geq 0$. For any such bin where $\sqrt{1 - q_x^2} < a_i$, we *know* that $\pi_i = 0$ for $q_y$ because $\|\boldsymbol{q}\| = 1$. Further, again due to the unit norm constraint, the conditional density for $q_y$ is in fact:

$$p(q_y|q_x) = \pi_k \frac{1}{\min(\sqrt{1 - q_x^2}, b_k) - a_k} \tag{2}$$

i.e., when $a_k < \sqrt{1 - q_x^2} < b_k$, the upper bound of the bin's uniform distribution is reduced. Similarly, $p(q_z|q_y, q_x) = \pi_k / (\min(\sqrt{1 - q_x^2 - q_y^2}, b_k) - a_k)$. The full density for $p(q_x, q_y, q_z)$ is thus:

$$p(q_x, q_y, q_z) = \pi_{q_x} \frac{N}{2} \pi_{q_y} \frac{1}{\omega_{q_y}} \pi_{q_z} \frac{1}{\omega_{q_z}} = \pi_{q_x} \pi_{q_y} \pi_{q_z} \frac{N}{2\omega_{q_y}\omega_{q_z}} \tag{3}$$

where $\pi_{q_c}$ is the mixture proportion for the bin containing component $q_c$ and $\omega_{q_c}$ is the potentially reduced width of the bin for $q_c$ used in Equation 2.

## 2.4 OPTIMIZING $p(\mathbf{q})$ AS A "QUATERNION LANGUAGE MODEL"

After incorporating the diluting factor from Section 2.2, the final density for $\boldsymbol{q} \in \widetilde{\mathbb{H}}_1$ is:

$$p(\boldsymbol{q}) = \pi_{q_x} \pi_{q_y} \pi_{q_z} \frac{Nq_w}{2\omega_{q_y}\omega_{q_z}} \tag{4}$$

As previously demonstrated by Alcorn & Nguyen (2021b), models using mixtures of uniform distributions that partition some domain can be optimized solely using a "language model loss". Here, the negative log-likelihood (NLL) for a dataset $\mathcal{X}$ is (where $d$ is the index for a sample):

$$\mathcal{L} = -\sum_{d=1}^{|\mathcal{X}|} \ln \pi_{q_{d,x}} + \ln \pi_{q_{d,y}} + \ln \pi_{q_{d,z}} + \ln \frac{Nq_{d,w}}{2\omega_{q_{d,y}}\omega_{q_{d,z}}} \tag{5}$$

Notice that the last term is constant for a given dataset, i.e., it can be ignored during optimization. The final loss is then $\widehat{\mathcal{L}} = -\sum_{d=1}^{|\mathcal{X}|} \ln \pi_{q_{d,x}} + \ln \pi_{q_{d,y}} + \ln \pi_{q_{d,z}}$, which is exactly the loss of a three token autoregressive language model.

As a final point, it is worth noting that the last term in Equation 4 is bounded below such that $\frac{Nq_w}{2\omega_{q_y}\omega_{q_z}} \geq \frac{N^3 q_w}{8}$, i.e., for a given $\pi_{q_x} \pi_{q_y} \pi_{q_z}$, the likelihood increases *at least* cubically with the number of bins. For 50,257 bins (i.e., the size of GPT-2/3's vocabulary (Radford et al., 2019; Brown et al., 2020)), $N^3 = 1.26 \times 10^{14}$. In contrast, the likelihood for IPDF only scales linearly with the number of equivolumetric grid cells (the equivalent user-defined precision hyperparameter).

# 3 AQUAMAM

## 3.1 ARCHITECTURE

Here, I describe AQuaMaM (Figure 4), a Transformer[10] that performs pose estimation using the autoregressive quaternion framework described in Section 2. Specifically, AQuaMaM consists of a

---

[10]See Section A.3 for a brief introduction to Transformers.

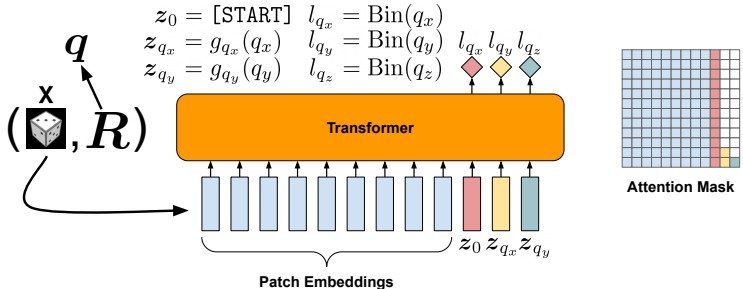

Figure 4: An overview of the AQuaMaM architecture. Given an image/rotation matrix pair $(\mathbf{X}, \boldsymbol{R})$, the image is first converted into a sequence of $P$ patch embeddings while the rotation matrix is converted into its unit quaternion representation $\boldsymbol{q} = [q_x, q_y, q_z, q_w]$. By restricting the unit quaternions to those with positive real components (which is allowed because $\boldsymbol{q}$ and $-\boldsymbol{q}$ encode the same rotation), $q_w$ becomes fully determined and does not need to be modeled. Next, each of the first two components $q_c$ of $\boldsymbol{q}$ is mapped to an embedding $z_{q_c}$ by a separate subnetwork $g_{q_c}$. The full input to the Transformer is thus a sequence consisting of the $P$ patch embeddings, a special [START] embedding $z_0$, and the two unit quaternion embeddings. The labels $l_{q_x}$, $l_{q_y}$, and $l_{q_z}$ are generated by assigning $q_x$, $q_y$, and $q_z$ to one of $N$ labels through a binning function Bin. Using a partially causal attention mask, AQuaMaM models the conditional distribution $p(q_x, q_y, q_z | \mathbf{X})$ *autoregressively*, i.e., $p(q_x, q_y, q_z | \mathbf{X}) = p(q_x | \mathbf{X}) p(q_y | q_x, \mathbf{X}) p(q_z | q_x, q_y, \mathbf{X})$ where each component is modeled as a mixture of uniform distributions that partition the component's geometrically constrained domain. Because minimizing the loss of a mixture of uniform distributions is equivalent (up to a constant) to minimizing the classification loss over the bins, AQuaMaM is trained as a "quaternion language model".

Vision Transformer (ViT) (Dosovitskiy et al., 2021) with a few modifications. As with ViT, AQuaMaM tokenizes an input image $\mathbf{X}$ by splitting it into $P$ patches that are then passed through a linear layer to obtain $P$ $d_{\text{model}}$-dimensional embeddings. Three additional embeddings are then appended to this sequence: (1) a [START] embedding $z_0$ (similar to the [START] embedding used in language models) and (2) two embeddings $z_{q_x}$ and $z_{q_x}$ corresponding to the first two components of $\boldsymbol{q}$. Each $z_{q_c}$ is obtained by passing $q_c$ through a subnetwork $g_{q_c}$ consisting of a NeRF-like positional encoding function (Vaswani et al., 2017; Mildenhall et al., 2020; Tancik et al., 2020) followed by a multilayer perceptron (MLP), i.e., the size of the input for the MLPs is $1 + 2 \times L$ where $L$ is the number of positional encoding functions.

Once positional embeddings have been added to the input sequence, the full sequence is then fed into a Transformer along with a partially causal attention mask (Alcorn & Nguyen, 2021a;b), which allows AQuaMaM to autoregressively model the conditional distribution $p(q_x, q_y, q_z | \mathbf{X})$ using the density from Equation 4. The transformed $z_0$, $z_{q_x}$, and $z_{q_y}$ embeddings—$\hat{z}_0$, $\hat{z}_{q_x}$, and $\hat{z}_{q_y}$—are used to assign a probability to the labels for $q_x$, $q_y$, and $q_z$, e.g., $\pi_{q_x} = h(\hat{z}_0)[l_{q_x}]$ where $h$ is the classification head and $l_{q_x} = \text{Bin}(q_x)$ where Bin is the binning function.[11] Notably, AQuaMaM somewhat sidesteps the curse of dimensionality by effectively learning three different models with shared parameters: $p(q_x | \mathbf{X})$, $p(q_y | q_x, \mathbf{X})$, and $p(q_z | q_y, q_x, \mathbf{X})$, each of which only needs to classify $N$ bins, as opposed to learning a single model that must classify on the order of $N^3$ bins.

### 3.2 GENERATING A SINGLE PREDICTION

While AQuaMaM *could* generate a single prediction by optimizing $\boldsymbol{q}$ under the full density (Equation 4)—similar to what was done in Murphy et al. (2021)—predictions obtained via optimization are slow. An alternative strategy is to embrace AQuaMaM as a "quaternion language model" and use a greedy search to return a "quaternion sentence" as a prediction. While sentences corresponding to regions where $q_w \approx 0$ will contain a wider range of compatible rotations, with a sufficiently large $N$, the range of rotations can be kept reasonably small. For example, when $N = 500$ (as in the AQuaMaM die experiment; see Section 4.2), if $q_x \in [0.996, 1]$, then $q_w \in [0, \sqrt{1 - 0.996^2}]$, and the geodesic distance between the rotations corresponding to the quaternions $\boldsymbol{q}_1 = [0.996, 0, 0, \sqrt{1 - 0.996^2}]$ and $\boldsymbol{q}_2 = [1, 0, 0, 0]$ is 10.3°. Using the analogous numbers

---

[11]PyTorch pseudocode for AQuaMaM's forward pass can be found in Listing 4 in Section A.6.

for $N = 50,257$ (as in the AQuaMaM toy experiment; see Section 4.1), the geodesic distance between $\boldsymbol{q}_1$ and $\boldsymbol{q}_2$ is 1.0°.

Generating a quaternion sentence with AQuaMaM requires three passes through the network. However, the last two passes can be considerably optimized by noting that the first $P + 1$ tokens do not depend on the last two tokens due to the partially causal attention mask. A naive decoding strategy where a full (padded) sequence is passed through AQuaMaM three times has an attention complexity of $O(3(P + 3)^2)$. By only passing the first $P + 1$ tokens through AQuaMaM and caching the attention outputs, the complexity of the attention operations when decoding reduces to $O((P + 1)^2 + (P + 2) + (P + 3))$. Empirically, I found that using this caching strategy increased the prediction throughput by approximately twofold.

## 4 EXPERIMENTS AND RESULTS

Compared to the standard datasets used for computer vision tasks like image classification (Deng et al., 2009) and object detection (Lin et al., 2014)—which contain millions of labeled samples—many commonly used pose estimation datasets are relatively small, with PASCAL3D+ (Xiang et al., 2014) consisting of $\sim$20,000 training images[12] and T-LESS (Hodan et al., 2017) consisting of $\sim$38,000 training images. The SYMSOL I and II datasets created by Murphy et al. (2021) consisted of 100,000 renders for each of eight different objects, but the publicly released versions only contain 50,000 renders per object.[13] In addition to their relatively small sizes, these different pose datasets have the additional complication of including multiple categories of objects. The "canonical" pose for a category of objects is arbitrary, so it is not entirely clear whether combining categories of objects makes pose estimation harder or easier for a learning algorithm (a jointly trained model essentially needs to do *both* classification and pose estimation).[14] Therefore, to investigate the properties of AQuaMaM in moderately large data regimes where Transformers excel, I trained AQuaMaM on two constructed datasets for pose estimation, an "infinite" toy dataset (Section 4.1) and a dataset of 500,000 renders of a die (Section 4.2). Because Murphy et al. (2021) found that IPDF outperformed other multimodal approaches by a wide margin in their distribution estimation task, I only compare to an IPDF baseline here.

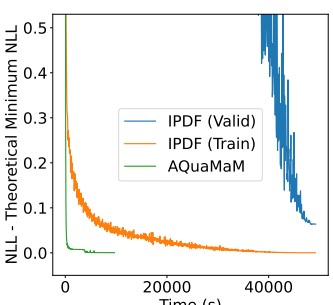

Figure 5: On the infinite toy dataset, AQuaMaM rapidly reached its theoretical minimum (classification) average negative log-likelihood (NLL). In contrast, IPDF never reached its theoretical minimum validation NLL, despite converging to its training theoretical minimum.

### 4.1 TOY EXPERIMENT

The infinite toy dataset consisted of six categories of viewpoints indexed $i = 0, 1, \ldots, 5$. I associated each viewpoint $i$ with $2^i$ randomly sampled rotation matrices $\mathcal{R}_i = \{\boldsymbol{R}_j\}_1^{2^i}$, i.e., each viewpoint had a different number of rotation modes reflecting its level of ambiguity. I defined the true data-generating process as a hierarchical model such that a sample was generated by first randomly selecting a viewpoint from a uniform distribution over $i$, and then randomly selecting a rotation from a uniform distribution over $\mathcal{R}_i$. Given this hierarchical model, calculating the average log-likelihood (LL) in the limit of infinite evaluation test samples simply involves replicating each $(i, \boldsymbol{R}_j)$ pair $2^{5-i}$ times and then averaging the LLs for each sample. Importantly, because the infinite dataset has identical training and test distributions, it is impossible for a model to overfit the dataset.

---

[12]Using the standard filtering pipeline where occluded and truncated images are excluded.

[13]The publicly released dataset can be found at: `https://www.tensorflow.org/datasets/catalog/symmetric_solids`.

[14]Further confusing the issue is the fact that many authors augment the PASCAL3D+ dataset with synthetic data, which makes it difficult to separate the influence of the architecture vs. the data augmentations when evaluating model performance.

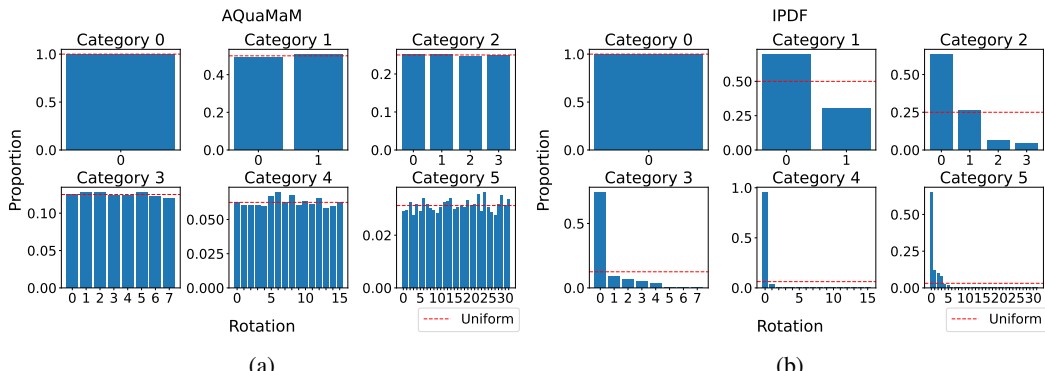

Figure 6: (a) The proportions of sampled rotations from the AQuaMaM model trained on the infinite toy dataset closely approximate the expected uniform distributions. (b) In contrast, despite approaching its theoretical minimum log-likelihood during training (Figure 5), the proportions of sampled rotations from the IPDF model drastically diverge from the expected uniform distributions.

Using this toy dataset, I trained an AQuaMaM model with $N = 50{,}257$.[15] Because I was only interested in the distribution learning aspects of the model for the toy dataset, the category for each training sample was mapped to a $P d_{\text{model}}$-dimensional embedding that was converted into a sequence of $P$ $d_{\text{model}}$-dimensional patch embeddings. I also trained an IPDF baseline using the same hyperparameters and learning rate schedule from Murphy et al. (2021) for the SYMSOL I dataset, except that I increased the number of iterations from 100,000 to 300,000 after observing that the model was still improving at the end of 100,000 updates. Like the AQuaMaM model, I replaced the image processing component of IPDF, a ResNet-50 (He et al., 2015), with $d_{\text{ResNet-50}}$-dimensional embeddings where $d_{\text{ResNet-50}} = 2{,}048$. In total, the AQuaMaM model had 3,510,609 parameters and the IPDF model had 748,545 parameters; however, 93% of AQuaMaM's parameters were contained in the final classification layer, with only 243,904 being found in the rest of the model.

The final average LL for AQuaMaM on the toy dataset was 27.12. In comparison, IPDF reached an average LL of 12.32 using the 2.4 million equivolumetric grid from Murphy et al. (2021). Note that the *theoretical maximum* LL for IPDF when using a 2.4 million grid is 12.38, which AQuaMaM comfortably surpassed. In fact, IPDF would need to use a grid of nearly six *trillion* cells for it to even be *theoretically* possible for it to match the LL of AQuaMaM.

Figure 6 shows the results of generating 40,000 samples from the hierarchical model while using AQuaMaM and IPDF to specify the rotation distributions for each sampled viewpoint.[16] For each sampled rotation, the rotation was assigned to the ground truth rotation mode from the sampled viewpoint with the smallest geodesic distance. For AQuaMaM, the average distance was 0.04°, while the average distance for IPDF was 0.84°. Further, the sampled rotations for AQuaMaM closely matched the true uniform distributions for each viewpoint. In contrast, IPDF's sampled distribution drastically diverged from the true data distribution despite its seemingly strong evaluation LL relative to its theoretical maximum. Notably, of the 40,000 quaternion sentences generated by AQuaMaM, only 23 (0.06%) were not from the true distribution.

## 4.2 DIE EXPERIMENT

The die dataset consisted of 520,000 renders[17] of a die (the same seen in Figure 1) where the die was randomly rotated to generate each image, and the dataset was split into training/validation/test set sizes of 500,000/10,000/10,000. Similar to the SYMSOL II dataset from Murphy et al. (2021), different viewpoints of the die have differing levels of ambiguity, which makes the die dataset appropriate for exploring how AQuaMaM handles uncertainty. Note that, while 500,000 samples is a moderately large training set, for $N = 500$, there are over 65 million unique token sequences

---

[15]Full training details for both experiments can be found in Section A.5.

[16]See Section A.4 for additional details.

[17]The die renders were generated using the renderer from Alcorn et al. (2019) and the 3D die model from: `https://github.com/garykac/3d-cubes`.

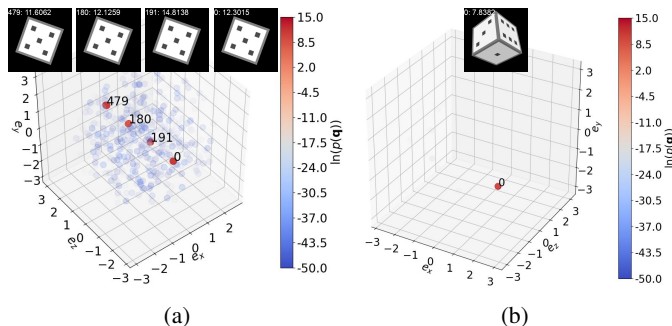

(a)                                                    (b)

Figure 7: AQuaMaM effectively models the uncertainty of different viewpoints for the die object. In the plots, each point corresponds to a rotation vector $\boldsymbol{\theta} = \theta\boldsymbol{e}$, i.e., all rotation vectors are contained within a ball of radius $\pi$. Both the color and the transparency of each point are a function of the log density $\ln(p(\boldsymbol{q}))$ with redder, more opaque points representing more likely rotations and bluer, more transparent points representing less likely rotations. 1,000 rotations are depicted in each plot—one rotation (labeled 0) corresponds to the ground truth, 499 rotations were obtained by sampling from the AQuaMaM distribution, and the remaining 500 rotations were randomly selected from the uniform distribution on $\mathbf{SO}(3)$. Each image of a die corresponds to one of the points in its respective plot. The number before the colon indicates the associated annotated point, and the number after the colon is the $\ln(p(\boldsymbol{q}))$ for that pose. (a) Because the five side of the die is facing the camera in the die's default pose, all rotations showing only the five occur along the $(0,0,1)$ axis, which is reflected by the locations of the red points in the plot. Additionally, there are four high probability regions along the axis corresponding to the four rotations that could produce an identical image (see Figure 8 in Section A.8 for a plot of rotations sampled from this density). (b) For this unambiguous viewpoint, almost all of the probability is concentrated at the correct rotation, which causes the other points to fade entirely from view (the *maximum* $\ln(p(\boldsymbol{q}))$ among the random rotations was -74.7).

covering $\mathbf{SO}(3)$,[18] and only 135 of the 10,000 token sequences in the test set were also found in the training set, i.e., AQuaMaM must still be able to generalize. I trained a $\sim$20 million parameter AQuaMaM with $N = 500$ on the die dataset in addition to a $\sim$26 million parameter IPDF baseline that was identical to the model trained on the SYMSOL I dataset in Murphy et al. (2021).

The average LL for AQuaMaM on the die dataset was 14.01 compared to 12.29 for IPDF, i.e., IPDF would need to use a grid of over 12 million cells for it to even be theoretically possible for it to match the average LL of AQuaMaM. When generating single predictions, AQuaMaM had a mean prediction error of 4.32° compared to 4.57° for IPDF, i.e., a 5.5% improvement.[19] In addition to reaching a higher average LL and a lower average prediction error, both the evaluation and prediction throughput for AQuaMaM were over 52× faster than IPDF on a single GPU. Lastly, as can be seen in Figure 7, AQuaMaM effectively models the uncertainty of viewpoints with differing levels of ambiguity. Figure 7 uses a novel visualization method where low probability rotations are more transparent, which allows all of the dimensions describing the rotation to be displayed, in contrast to the visualization method introduced by Murphy et al. (2021).[20]

## 5    CONCLUSION

In this paper, I have shown that a Transformer trained using an autoregressive "quaternion language model" framework is highly effective for learning complex distributions on the $\mathbf{SO}(3)$ manifold. Compared to IPDF, AQuaMaM makes more accurate predictions at a much faster throughput and produces far more accurate sampling distributions. The autoregressive factoring of the quaternion could be applied to other multidimensional datasets where high levels of precision are desirable, e.g., modeling the trajectory of a basketball as in Alcorn & Nguyen (2021a), which was partial inspiration for this work.

---

[18]$500^3 \times \left(\frac{4\pi}{3} \div 2^3\right) \approx 65,000,000$, i.e., the fraction of grid cells in a cube with a side of two that fall inside the unit sphere.

[19]See Section A.7 for additional details.

[20]Additional experiments for a cylinder dataset and a mixture of Gaussians design can be found in Section A.9.

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

## A    APPENDIX

### A.1    TRANSFORMING A DENSITY ON THE UNIT DISK TO A DENSITY ON THE UNIT HEMISPHERE

Here, I briefly walk through the diluting procedure for a 2D example that is directly analogous to $\widetilde{\mathbb{H}}_1$ and $B^3$. Consider the task of approximating a uniform distribution on the unit hemisphere $\widetilde{S}^2 = \{v \in \mathbb{R}^3 : \|v\| = 1 \text{ and } v_z > 0\}$ by modeling the projected coordinates of $v \in \widetilde{S}^2$ as a mixture of uniform distributions on the unit disk $B^2 = \{v \in \mathbb{R}^2 : \|v\| < 1\}$ (Figure 2). Note that $f(x, y) = [x, y, z]$ and:

$$
J = \begin{bmatrix} 1 & 0 \\ 0 & 1 \\ \frac{-x}{z} & \frac{-y}{z} \end{bmatrix}
$$

where $z = \sqrt{1 - x^2 - y^2}$. The columns of $J$ define a parallelogram tangent to the hemisphere at $(x, y, z)$ with an area equal to the square root of the sum of the squared $2 \times 2$ minors in $J^\top$,[21][22] i.e.:

$$
a = \sqrt{\begin{vmatrix} 0 & 1 \\ \frac{-x}{z} & \frac{-y}{z} \end{vmatrix}^2 + \begin{vmatrix} 1 & 0 \\ \frac{-x}{z} & \frac{-y}{z} \end{vmatrix}^2 + \begin{vmatrix} 1 & 0 \\ 0 & 1 \end{vmatrix}^2} = \sqrt{\frac{x^2}{z^2} + \frac{y^2}{z^2} + 1} = \sqrt{\frac{x^2 + y^2 + z^2}{z^2}} = \frac{1}{z}
$$

We can now calculate $p(x, y, z)$ as $p(x, y, z) = \frac{p(x,y)}{a} = p(x, y)z$. Intuitively, $z = 1$ corresponds to the top of the hemisphere where the tangent parallelogram is parallel to $B^2$ and there is no expansion/dilution. In contrast, as $z \to 0$, the tangent parallelogram becomes increasingly "stretched" leading to greater dilution of $p(x, y)$.

The recipe for calculating $s_q$ is nearly identical, with:

$$
J = \begin{bmatrix} 1 & 0 & 0 \\ 0 & 1 & 0 \\ 0 & 0 & 1 \\ \frac{-q_x}{q_w} & \frac{-q_y}{q_w} & \frac{-q_z}{q_w} \end{bmatrix}
$$

and $3 \times 3$ minors being used in place of $2 \times 2$ minors. As a result, $s_q = \frac{1}{q_w}$, i.e., the diluting factor for AQuaMaM is $q_w$.

### A.2    DEFINING "STRICTLY ILLEGAL BINS"

The definition of "strictly illegal bins" used by AQuaMaM is best motivated by an example. Consider a model that was trained on a dataset consisting of a single rotation. Python pseudocode for a naive sampling algorithm can be found in Listing 1 where the `get_probs_naive` function assigns a probability of zero to any bins where all of the values would violate the unit norm constraint given the components of $q$ generated up to that point. This naive sampling procedure will generate sequences of bins that were not found in the training dataset. For example, assume the $q$ for the single rotation had $q_x = 0.42$ and $q_y = 0.901$ and that $N = 20$ for the model. Given a globally optimal model, $\hat{q}_x$ will be sampled from $[0.4, 0.5]$; however, if $\hat{q}_x > \sqrt{1 - 0.9^2} \approx 0.436$, then `get_probs_naive` will assign zero probability to the bin $[0.9, 1.0]$.

```
1  def naive_sample_q(AQuaMaM):
2      q_cs = []
3      for i in range(3):
4          bin_probs = AQuaMaM.get_probs_naive(q_cs)
5          l_q_c = categorical(bin_probs)
```

---

[21]This procedure resembles the "formal determinant" method for computing the cross product: `https://en.wikipedia.org/wiki/Cross_product#Matrix_notation`.

[22]The area of the parallelogram can also be computed by taking the square root of the determinant of the Gram matrix $G = J^\top J$.

```
6          (lower, upper) = AQuaMaM.bin_edges[l_q_c]
7          q_cs.append(constrained_uniform(lower, upper, q_cs))
8
9      (q_x, q_y, q_z) = q_cs
10     q_w = sqrt(1 - q_x**2 - q_y**2 - q_z**2)
11     return (q_x, q_y, q_z, q_w)
```

Listing 1: Python pseudocode for naively generating a unit quaternion sample with AQuaMaM.

To avoid this issue, AQuaMaM defines "strictly illegal bins" as those that would violate the unit norm constraint given the components of $q$ generated up to that point *when using the minimum magnitudes of their corresponding bins*. Effectively, during training AQuaMaM must *learn* to assign zero probability to bins bordering "edge bins" rather than being *told* that they have zero probability (which is the case for strictly illegal bins). Python pseudocode for the final sampling algorithm can be found in Listing 2. In summary, the sampling procedure involves first generating a sequence of bins, and then sampling from the region defined by the edges of those bins using rejection sampling. When viewed as a generative model, AQuaMaM can be interpreted as modeling noisy measurements of unit quaternions.

```
1  def sample_q(AQuaMaM):
2      # The q_c_hats are used for generating the bins, but are not the
       final q_cs.
3      q_c_hats = []
4      cell_edges = []
5      for i in range(3):
6          bin_probs = AQuaMaM.get_probs(q_c_hats)
7          l_q_c = categorical(bin_probs)
8          (lower, upper) = AQuaMaM.bin_edges[l_q_c]
9          q_c_hats.append(uniform(lower, upper))
10         cell_edges.append((lower, upper))
11
12     (q_x, q_y, q_z) = rejection_sample(cell_edges)
13     q_w = sqrt(1 - q_x**2 - q_y**2 - q_z**2)
14     return (q_x, q_y, q_z, q_w)
```

Listing 2: Python pseudocode for generating a unit quaternion sample with AQuaMaM.

### A.3 BACKGROUND ON TRANSFORMERS

In this section, I briefly describe the Transformer (Vaswani et al., 2017) architecture at a *high* level, and define the Transformer-specific terms used in the main text. Readers who seek a deeper understanding of Transformers are encouraged to explore the following excellent pedagogical materials:

1. The Illustrated Transformer by Jay Alammar: `https://jalammar.github.io/illustrated-transformer/`

2. The Annotated Transformer by Sasha Rush, Austin Huang, Suraj Subramanian, Jonathan Sum, Khalid Almubarak, and Stella Biderman: `http://nlp.seas.harvard.edu/annotated-transformer/`

3. Attention? Attention! by Lilian Weng `https://lilianweng.github.io/posts/2018-06-24-attention/`

Transformers are a class of neural networks that are capable of processing sequential data *without being explicitly sequential*. To understand what that means, it is instructive to contrast Transformers with recurrent neural networks (RNNs), which *are* explicitly sequential. Both Transformers and RNNs take as input a sequence of $N$ $d$-dimensional vectors and output a sequence of $N$ $d$-dimensional vectors. During both training and testing, RNNs operate by iteratively applying the same neural network to each element of a sequence and an evolving hidden state. In contrast, during training, Transformers process an entire sequence *in parallel*.

To accomplish this parallelism, Transformers use the attention mechanism (Graves, 2013; Graves et al., 2014; Weston et al., 2015; Bahdanau et al., 2015).[23] Intuitively, the attention mechanism is

---

[23]"Attention is all [they] need" - Vaswani et al. (2017).

a function that tells a neural network how much to "focus" on the different elements of a sequence when processing a specific element of that sequence. The pure attention approach to sequence modeling provides two main benefits: (1) it can greatly speed up training because sequences are processed in parallel through the network, and (2) long-term dependencies are much easier to model because distant elements of the sequence do not need to communicate through many neural network layers (which *is* the case for RNNs).

While the attention mechanism is extremely powerful, it is also permutation invariant. As a result, Transformers must encode positional information and conditional dependencies through other means, i.e., with position embeddings and attention masks. Consider the following sequences of word embeddings:[24]

$$\mathcal{W}_1 = [\boldsymbol{w}_{\texttt{[START]}}, \boldsymbol{w}_{\text{the}}, \boldsymbol{w}_{\text{cat}}, \boldsymbol{w}_{\text{in}}, \boldsymbol{w}_{\text{the}}, \boldsymbol{w}_{\text{hat}}]$$
$$\mathcal{W}_2 = [\boldsymbol{w}_{\texttt{[START]}}, \boldsymbol{w}_{\text{in}}, \boldsymbol{w}_{\text{the}}, \boldsymbol{w}_{\text{cat}}, \boldsymbol{w}_{\text{the}}, \boldsymbol{w}_{\text{hat}}]$$

Because the attention mechanism is permutation invariant, $\mathcal{W}_1$ and $\mathcal{W}_2$ look identical to a Transformer. Therefore, to encode the *order* of the word embeddings, the Transformer adds a **position embedding** $\boldsymbol{t}_i$ to each word embedding where $i$ indicates the word embedding's position in the sequence. As a result, $\mathcal{W}_1$ becomes:

$$\widetilde{\mathcal{W}}_1 = [\boldsymbol{w}_{\texttt{[START]}} + \boldsymbol{t}_1, \boldsymbol{w}_{\text{the}} + \boldsymbol{t}_2, \boldsymbol{w}_{\text{cat}} + \boldsymbol{t}_3, \boldsymbol{w}_{\text{in}} + \boldsymbol{t}_4, \boldsymbol{w}_{\text{the}} + \boldsymbol{t}_5, \boldsymbol{w}_{\text{hat}} + \boldsymbol{t}_6]$$

and the attention mechanism would be applied to $\widetilde{\mathcal{W}}_1$ and $\widetilde{\mathcal{W}}_2$.

In order to fully understand how Transformers model conditional dependencies, additional technical details about the attention mechanism are necessary. The basic **attention mechanism** in Transformers consists of two architectural pieces: (1) a query/key/value function $\phi$ and (2) a score function $\psi$. The **query/key/value function** $\phi$ independently maps each element of a (position encoded) sequence $\widetilde{\mathcal{W}}[i]$ to a query, key, and value vector, i.e.:

$$[\boldsymbol{q}_i, \boldsymbol{k}_i, \boldsymbol{v}_i] = \phi(\widetilde{\mathcal{W}}[i])$$

The **score function** $\psi$ takes a query vector $\boldsymbol{q}_i$ and a key vector $\boldsymbol{k}_j$ as input and outputs a scalar score, i.e.:

$$s_{i,j} = \psi(\boldsymbol{q}_i, \boldsymbol{k}_j)$$

The score matrix $\boldsymbol{\Psi}$ is thus an $N \times N$ matrix where $\boldsymbol{\Psi}[i, j] = s_{i,j}$. The final step of the attention mechanism involves calculating a new vector for each element in the sequence:

$$\widehat{\mathcal{W}}[i] = \sum_{j=1}^{N} a_{i,j} \boldsymbol{v}_j$$

where:

$$a_{i,j} = \frac{e^{s_{i,j}}}{\sum_{k=1}^{N} e^{s_{i,k}}}$$

i.e., the **attention weights** $a_{i,j}$ are calculated by applying the softmax function to each row of $\boldsymbol{\Psi}$.

Now consider the common natural language processing task of "language modeling", i.e., learning $p(\mathcal{S})$ where $\mathcal{S}$ is a sentence with $N - 1$ words. A common way to factorize $p(\mathcal{S})$ is as:[25]

$$p(\mathcal{S}) = p(\mathcal{S}[1])p(\mathcal{S}[2]|\mathcal{S}[1]) \dots p(\texttt{[STOP]}|\mathcal{S}[N-1], \dots, \mathcal{S}[2], \mathcal{S}[1])$$

---

[24]$\boldsymbol{w}_{\texttt{[START]}}$ is a special embedding that is appended to the beginning of sequences in both Transformers and RNNs in situations where the first element of the sequence is being modeled.

[25][STOP] is a special token indicating the end of a sentence.

which is using the chain rule of probability. To enforce such conditional dependencies, Transformers use an **attention mask** $M$, which is an $N \times N$ matrix where $M[i,j] = 0$ if the Transformer is allowed to "look" at element $\widetilde{\mathcal{W}}[j]$ of the sequence when processing element $\widetilde{\mathcal{W}}[i]$, and $M[i,j] = -\infty$ if the "looking" is not allowed. Instead of applying the softmax function to each row of the raw score matrix $\mathbf{\Psi}$, Transformers apply the softmax function to each row of a *masked* score matrix $\widetilde{\mathbf{\Psi}} = \mathbf{\Psi} + M$. As a result, $a_{i,j} = 0$ when the Transformer is not allowed to "look" at element $\widetilde{\mathcal{W}}[j]$ of the sequence when processing element $\widetilde{\mathcal{W}}[i]$ (because $e^{-\infty} = 0$).

To model the chain rule factorization described earlier, $M$ takes the form of a strictly upper triangular matrix where all of the values above the diagonal are set to $-\infty$. Because a strictly upper triangular attention mask means each element in the sequence cannot be influenced by elements that occur *later* than that element in the sequence (i.e., the present is only affected by the past), this particular attention mask is referred to as a ***causal* attention mask**. Because AQuaMaM is modeling the distribution of $\widetilde{\mathbb{H}}_1$ *conditioned on an image*, the Transformer is allowed to "look" at *all* of the image patch embeddings when processing any single patch embedding, i.e., AQuaMaM uses a ***partially* causal attention mask**, which is depicted in Figure 4.

The full Transformer architecture consists of $L$ blocks where each block (indexed by $\ell$) first applies the masked attention mechanism to the full sequence with $\phi_\ell$ and $\psi_\ell$, and then passes each element of the resulting sequence through an MLP $f_\ell$, i.e.:

$$\ddot{\mathcal{W}}[i] = f_\ell(\widehat{\mathcal{W}}[i])$$

As previously mentioned, the pure attention approach to sequence modeling means Transformers can evaluate entire sequences in parallel, i.e., the likelihood of a full sequence can be calculated in a single forward pass. However, both making predictions and generating samples require $N$ forward passes through the Transformer because the elements of the sequence can only be "filled in" one by one. The naive Transformer **decoding** algorithm thus starts off with a sequence of [NULL] embeddings[26] following the [START] embedding, e.g.:

$$\mathcal{W} = \left[ \boldsymbol{w}_{\texttt{[START]}}, \boldsymbol{w}_{\texttt{[NULL]}}, \boldsymbol{w}_{\texttt{[NULL]}}, \boldsymbol{w}_{\texttt{[NULL]}}, \boldsymbol{w}_{\texttt{[NULL]}}, \boldsymbol{w}_{\texttt{[NULL]}} \right]$$

and then iteratively generates the words of the sentence. As discussed in Section 3.2, the caching strategy used by AQuaMaM is motivated by the fact that this naive decoding algorithm is computationally wasteful with regards to the attention mechanism.

### A.4 GENERATING SAMPLES

Generating samples with IPDF simply involves sampling from the probability distribution defined by IPDF over the equivolumetric grid. For AQuaMaM, the sampling procedure uses a conditional version of the algorithm found in Listing 2. Python pseudocode for the conditional AQuaMaM sampling procedure can be found in Listing 3.

```
def sample_q(AQuaMaM, image):
    # The q_c_hats are used for generating the bins, but are not the
    final q_cs.
    q_c_hats = []
    cell_edges = []
    for i in range(3):
        bin_probs = AQuaMaM.get_probs(image, q_c_hats)
        l_q_c = categorical(bin_probs)
        (lower, upper) = AQuaMaM.bin_edges[l_q_c]
        q_c_hats.append(uniform(lower, upper))
        cell_edges.append((lower, upper))

    (q_x, q_y, q_z) = rejection_sample(cell_edges)
    q_w = sqrt(1 - q_x**2 - q_y**2 - q_z**2)
```

---

[26]For example, a vector full of zeros, although the actual values are irrelevant.

```
14      return (q_x, q_y, q_z, q_w)
```

Listing 3: Python pseudocode for generating a unit quaternion sample conditioned on an image with AQuaMaM.

## A.5 AQuaMaM training details

### A.5.1 Toy experiment

For the toy dataset, the hyperparameters for the AQuaMaM model were $d_{\text{model}} = 64$, $P = 196$ (i.e., the number of $16 \times 16$ patches in a $224 \times 224$ image), eight attention heads, $d_{\text{ff}} = 256$ (the dimension of the inner feedforward layers of the Transformer), three Transformer layers, $L = 6$, $g_{q_c}$ MLPs with 16, 32, and 64 output nodes and Gaussian Error Linear Unit (GELU) nonlinearities (Hendrycks & Gimpel, 2016), and $N = 50,257$. I used the Adam optimizer (Kingma & Ba, 2015) with an initial learning rate of $10^{-4}$, $\beta_1 = 0.9$, $\beta_2 = 0.999$, and $\epsilon = 10^{-9}$ to update the model's parameters. The learning rate was reduced by half every time the validation loss did not improve for five consecutive epochs. The model was implemented in PyTorch and trained with a batch size of 128 on a single Tesla V100 GPU for ~2.5 hours (624 epochs) where each epoch consisted of 40,000 training samples, and the evaluation loss was used for early stopping.

### A.5.2 Die experiment

For the die dataset, the hyperparameters for the AQuaMaM model were $d_{\text{model}} = 512$, $P = 196$, eight attention heads, $d_{\text{ff}} = 2,048$, six Transformer layers, $L = 6$, $g_{q_c}$ MLPs with 128, 256, and 512 output nodes and GELU nonlinearities, and $N = 500$. In total, this AQuaMaM model had 20,000,756 parameters. The model was trained using an identical optimization strategy to the toy experiment, and was trained for ~2 days (121 epochs).

### A.6 THE AQUAMAM FORWARD PASS

```
1  # Shape values come from Section A.5.
2
3  # imgs has a shape of (128, 3, 224, 224).
4  # qs has a shape of (128, 4).
5  # self.z_0 has a shape of (1, 1, 512).
6  # self.pos_embed (the position embeddings) has a shape of (1, 199, 512).
7  # self.attn_mask has a shape of (199, 199).
8  # self.h = nn.Sequential(nn.LayerNorm(512), nn.Linear(512, 500)).
9  def forward(self, imgs, qs):
10     # patch_embeds has a shape of (128, 196, 512).
11     patch_embeds = self.patch_embed(imgs)
12
13     # z_0 has a shape of (128, 1, 512).
14     z_0 = self.z_0.expand(len(patch_embeds), -1, -1)
15     # z_q_x has a shape of (128, 1, 512).
16     z_q_x = self.g_q_x(qs[:, 0])
17     # z_q_y has a shape of (128, 1, 512).
18     z_q_y = self.g_q_y(qs[:, 1])
19
20     # x has a shape of (128, 199, 512), i.e., 199 = 196 + 3.
21     x = torch.cat([patch_embeds, z_0, z_q_x, z_q_y], dim=1)
22     x = x + self.pos_embed
23     # x still has a shape of (128, 199, 512).
24     x = self.transformer(x, self.attn_mask)
25
26     # x has a shape of (128, 3, 500).
27     x = self.h(x[:, -3:])
28
29     # See Section 2.3. x still has a shape of (128, 3, 500).
30     x = constrain_qs(qs, self.bins, x)
31
32     return x
```

Listing 4: PyTorch pseudocode for the AQuaMaM forward pass.

### A.7 GENERATING PREDICTIONS

Generating a prediction with IPDF simply involves taking the highest probability element from the probability distribution defined by IPDF over the equivolumetric grid. For AQuaMaM, generating a deterministic prediction is similar to the steps for generating a sample (See Listing 3), except the maximum probability bin is always selected and $q_c$ is set as the midpoint of the selected bin rather than sampled from it. Python pseudocode for the conditional AQuaMaM sampling procedure can be found in Listing 5.

```
1  def predict_q(AQuaMaM, image):
2      # The q_c_hats are used for selecting the bins, but are not the final
         q_cs.
3      q_c_hats = []
4      for i in range(3):
5          l_q_c = AQuaMaM.get_probs(image, q_c_hats).argmax()
6          (lower, upper) = AQuaMaM.bin_edges[l_q_c]
7          q_c_hats.append((lower + upper) / 2)
8
9      if norm(q_c_hats) <= 1:
10         q_cs = q_c_hats
11     else:
12         q_cs = normalize(q_c_hats)
13
14     q_w = sqrt(1 - q_x**2 - q_y**2 - q_z**2)
15     return (q_x, q_y, q_z, q_w)
```

Listing 5: Python pseudocode for deterministically predicting a unit quaternion given an image with AQuaMaM.

## A.8 SAMPLED ROTATIONS FOR FIGURE 10A

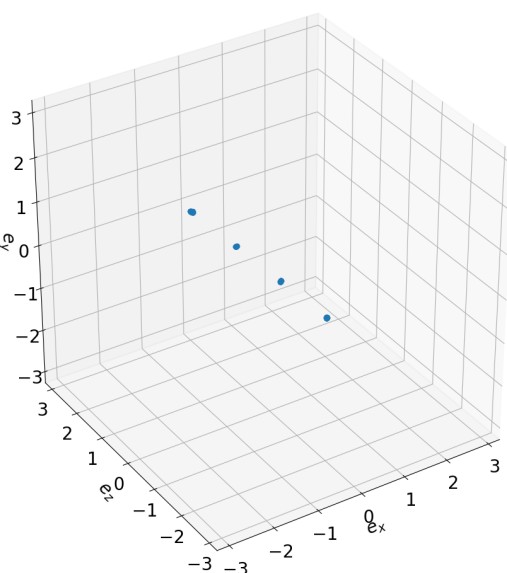

Figure 8: 1,000 sampled rotations from the AQuaMaM density shown in Figure 10a. Each of the equivalent modes is represented, and there are no points in incorrect regions.

## A.9 ADDITIONAL EXPERIMENTS

### A.9.1 CYLINDER EXPERIMENT

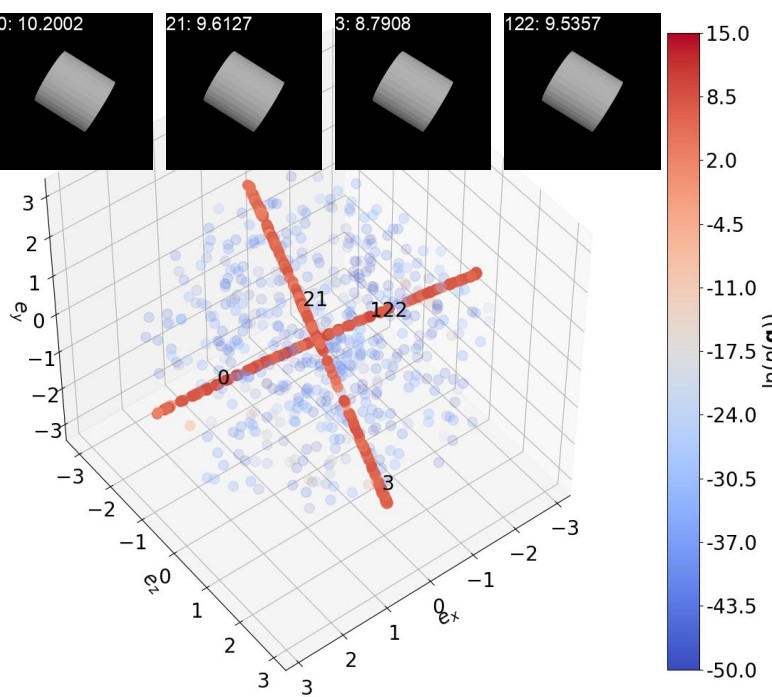

Figure 9: AQuaMaM is also capable of learning distributions for objects with continuous symmetries, as shown here for a viewpoint of the cylinder. 1,000 rotations are depicted in the plot—one rotation (labeled 0) corresponds to the ground truth, 499 rotations were obtained by sampling from the AQuaMaM distribution, and the remaining 500 rotations were randomly selected from the uniform distribution on $\mathbf{SO}(3)$. The cylinder's two axes of symmetry are clearly visible as high density curves in the rotation vector ball.

Following the same training setup as the die experiment (Section 4.2), I trained an AQuaMaM and IPDF model on a cylinder[27] dataset (notably, a cylinder is one of the objects in the SYMSOL I dataset). The average LL for AQuaMaM on the cylinder dataset was 7.24 compared to 5.94 for IPDF.[28] Figure 9 visualizes the distribution learned by AQuaMaM using the same visualization technique described in Figure 7.

### A.9.2 MIXTURE OF GAUSSIANS EXPERIMENT

This section describes a mixture of Gaussians (MoG) variant of AQuaMaM, AQuaMaM-MoG, which is conceptually similar to how (non-curved) densities are modeled in RNADE (Uria et al., 2013). As a preliminary, recall the change-of-variable technique:[29]

> Let $X$ be a continuous random variable with generic probability density function $f(x)$ defined over the support $c_1 < x < c_2$. And, let $Y = u(X)$ be an *invertible* function of $X$ with inverse function $X = v(Y)$. Then, using the **change-of-variable technique**, the probability density function of $Y$ is:

---

[27]The 3D cylinder model can be found at: `https://github.com/jlamarche/Old-Blog-Code/blob/master/Wavefront OBJ Loader/Models/Cylinder.obj`.

[28]The reported average LL on the cylinder dataset in Murphy et al. (2021) was 4.26, but they used a considerably smaller training dataset (100,000 vs. 500,000) and updated the model considerably fewer times during training (100,000 vs. 300,000).

[29]Taken verbatim from: `https://online.stat.psu.edu/stat414/lesson/22/22.2`.

$$f_Y(y) = f_X(v(y)) \times |v'(y)|$$

**defined over the support** $u(c_1) < y < u(c_2)$.

Let $q_c = \ell_{q_c} + (u_{q_c} - \ell_{q_c})\frac{1}{1+e^{-s_{q_c}}}$ where $\ell_{q_c}$ and $u_{q_c}$ define the lower and upper bounds for quaternion component $q_c$ (i.e., $\ell_{q_c} = -u_{q_c}$), and $s_{q_c}$ is distributed according to a MoG parameterized by AQuaMaM-MoG. In words, $s_{q_c}$ determines where $q_c$ falls in $(-u_{q_c}, u_{q_c})$ through the logistic function. Solving for $s_{q_c}$ (to get the equivalent of $v(y)$) gives:

$$q_c = -u_{q_c} + 2u_{q_c}\frac{1}{1+e^{-s_{q_c}}}$$

$$1 + e^{-s_{q_c}} = \frac{2u_{q_c}}{q_c + u_{q_c}}$$

$$e^{-s_{q_c}} = \frac{2u_{q_c}}{q_c + u_{q_c}} - 1$$

$$e^{-s_{q_c}} = \frac{2u_{q_c} - (q_c + u_{q_c})}{q_c + u_{q_c}}$$

$$e^{-s_{q_c}} = \frac{u_{q_c} - q_c}{q_c + u_{q_c}}$$

$$-s_{q_c} = \ln(u_{q_c} - q_c) - \ln(q_c + u_{q_c})$$

$$\ln(q_c + u_{q_c}) - \ln(u_{q_c} - q_c) = s_{q_c}$$

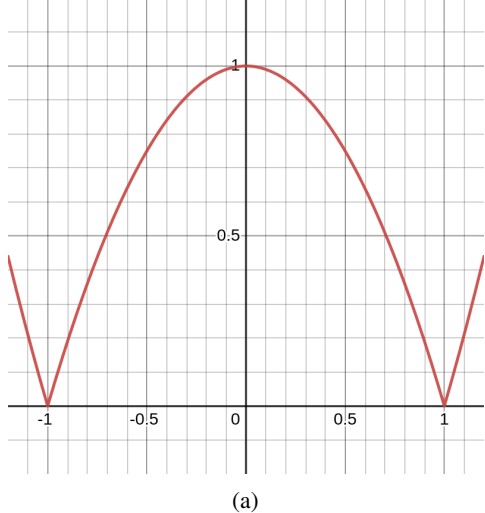
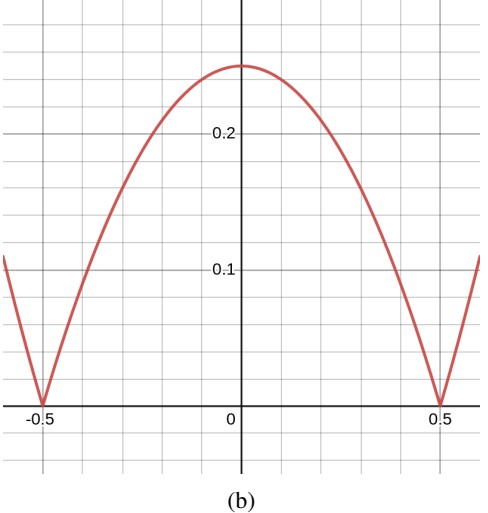

(a)                (b)

Figure 10: Plots of the denominator for the $|v'(y)|$ term in the change-of-variable formula when using a mixture of Gaussians approach with AQuaMaM. (a) $= |-q_c^2 + 1^2|$ and (b) $= |-q_c^2 + 0.5^2|$.

Differentiating with respect to $q_c$ (to get the equivalent of $v'(y)$) gives:

$$\frac{d}{dq_c}[\ln(q_c + u_{q_c}) - \ln(u_{q_c} - q_c)] = \frac{1}{q_c + u_{q_c}} + \frac{1}{u_{q_c} - q_c}$$

$$= \frac{u_{q_c} - q_c + q_c + u_{q_c}}{(q_c + u_{q_c})(u_{q_c} - q_c)}$$

$$= \frac{2u_{q_c}}{u_{q_c}^2 - q_c^2}$$

Note that the minimum of $|u_{q_c}^2 - q_c^2|$ is zero and occurs when $q_c = \pm u_{q_c}$ (see Figure 10), i.e., $|v'(y)| \to \infty$ as $q_c \to \pm u_{q_c}$.

The full AQuaMaM-MoG density is thus:

$$p(\boldsymbol{q}) = \pi_{s_{q_x}} \left| \frac{2}{u_{q_x}^2 - q_x^2} \right| \pi_{s_{q_y}} \left| \frac{2u_{q_y}}{u_{q_y}^2 - q_y^2} \right| \pi_{s_{q_z}} \left| \frac{2u_{q_z}}{u_{q_z}^2 - q_z^2} \right| q_w \tag{6}$$

where $\pi_{s_{q_c}}$ is the density assigned to $s_{q_c}$ by the MoG parameterized by AQuaMaM-MoG. Because the change-of-variable terms are constant for a given dataset, the training loss for AQuaMaM-MoG is:

$$\mathcal{L} = -\sum_{d=1}^{|\mathcal{X}|} \ln \pi_{d,s_{q_x}} + \ln \pi_{d,s_{q_y}} + \ln \pi_{d,s_{q_z}} \tag{7}$$

I trained an AQuaMaM-MoG model on the toy dataset described in Section 4.1 using 512 mixture components and otherwise identical hyperparameters as described in Section A.5.1. The LL on the test set for AQuaMaM-MoG was 10.52, which is not only far worse than AQuaMaM (27.12), but is also considerably worse than IPDF (12.32). Further, the sampling distribution of AQuaMaM-MoG is far from the true data distribution, as can be seen in Figure 11.

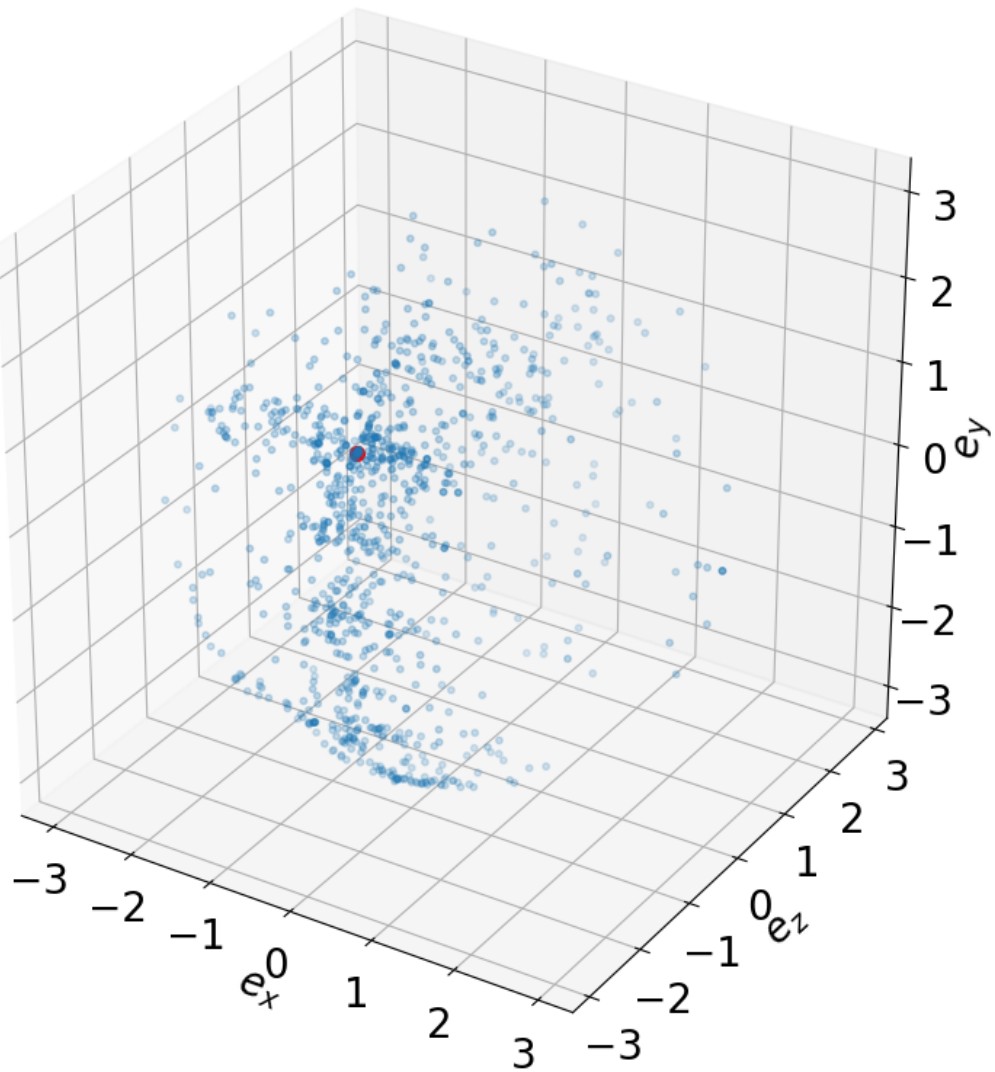

Figure 11: Rotations sampled from AQuaMaM-MoG for the unimodal viewpoint in the toy dataset are often far from the true rotation (the red point).

