# OpenReview forum: "AQuaMaM: An Autoregressive, Quaternion Manifold Model for Rapidly Estimating Complex SO(3) Distributions"
_ICLR.cc/2023/Conference — Submitted to ICLR 2023_

### Official Review · Reviewer_WNVz · 2022-10-24

**Confidence:** 4
**Correctness:** 3
**Technical Novelty And Significance:** 2
**Empirical Novelty And Significance:** 3
**Recommendation:** 5

**Clarity, Quality, Novelty And Reproducibility:**

The choice to use a mixture of uniform distributions was surprising and a bit odd to me.  One would expect that problems like the ones examined would exhibit some amount of local smoothness that is not easily captured with the mixture of uniform construction.  Another option could have been, for example to use a mixture of Gaussians for each conditional (as, for example, in RNADE MOG https://www.pure.ed.ac.uk/ws/portalfiles/portal/11874029/1068.pdf).  Can the authors speak to this choice? Have they considered a mixture of Gaussians for the conditionals with appropriate rescaling (e.g. via softmax) to the constrained space?

A chief advantage of the approach is said to be the computational speed-up relative to Murphy 2021.  Is computation cost a limiting factor in the applications to which this method is applicable?

The authors describe their method and characterize it relative to the work of Murphy 2021.  Why do the authors choose this as the base of comparison? I was not previously familiar with this approach so was surprised to see it treated as a gold-standard baseline to beat.  Without a compelling explanation for the choice of this baseline the empirical results do not stand on their own as impressive.

There has been quite a lot of recent work in this space.  Can the authors comment on advantages and disadvantages to previous work including:

De Bortoli, Valentin, et al. "Riemannian score-based generative modeling." arXiv preprint arXiv:2202.02763 (2022).

Falorsi, L., de Haan, P., Davidson, T. R., and Forré, P. (2019). Reparameterizing distributions on lie groups. In International Conference on Artificial Intelligence and Statistics, pages 3244–3253.

Leach, Adam, et al. "Denoising Diffusion Probabilistic Models on SO (3) for Rotational Alignment." ICLR 2022 Workshop on Geometrical and Topological Representation Learning. 2022.

---

The writing in the paper is generally very clear.

**Strength And Weaknesses:**

Strengths:
* The method is relatively simple and very computationally efficient as compared to the approach of Murphy.
* The experimental results in figure 7 are illustrative.  This is a very nice choice of visual.

Weaknesses:
* The method is not very clearly motivated relative to other possibilities.  E.g. why use quaternions as the underlying representation on which to apply the autoregressive decomposition? And why use a mixture of uniforms rather than something else with a computable likelihood?
* The empirical component of the validation is very limited.  The authors could strength the paper with comparisons to additional methods.
* The experimental results in figure 7 A appear to be a negative result.  There is much probability mass spread throughout the space, not just on the four possible orientations given the top view of the “5”.


**Summary Of The Paper:**

The paper describes an approach to learning distributions on the SO(3) manifold via an autoregressive factorization of the distribution on quaternions, wherein each conditional and the first marginal are approximated by a mixture of uniform distributions.

The paper provides a comparison to the prior work of Murphy 2021, and shows some empirical advantages over that approach.

**Summary Of The Review:**

The paper describes a new approach for modeling distributions on the SO3 manifold.  I recommend rejection because of little motivation for the choice of the method relative to other possibilities, and because of a lack of substantial positive empirical results to demonstrate that the method performs well.

Update: I have changed my score from 3 to 5.

---

> ### Author Response · Authors · 2022-11-10
> **Response to Reviewer WNVz**
>
> Thank you for your review. I’m happy you agree AQuaMaM has clear advantages over IPDF, and I appreciate you highlighting the interpretability properties of my data visualization technique.
>
> >The experimental results in figure 7 A appear to be a negative result. There is much probability mass spread throughout the space, not just on the four possible orientations given the top view of the “5”.
>
> I mentioned in the caption of Figure 7 (and it's indicated on the colorbar) that the points are shaded according to their *log densities* (i.e., *not* their probability masses). The shading scheme was chosen to ensure that the reader could see the distribution of randomly sampled rotations. The *minimum* log density of the four highlighted red points is 2.4235, which corresponds to a density of $e^{2.4235} = 11.29$. The *maximum* log density of the randomly sampled rotations was -21.1295, which corresponds to a density of $e^{-21.1295} = 6.6 \times 10^{-10}$, i.e., AQuaMaM is dedicating a vanishingly small amount of probability to the random rotations, which is why it’s not a negative result.
>
> >The method is not very clearly motivated relative to other possibilities. E.g. why use quaternions as the underlying representation on which to apply the autoregressive decomposition?
>
> >The choice to use a mixture of uniform distributions was surprising and a bit odd to me.
>
> Unit quaternions are a widely used formalism for encoding rotations (as emphasized by Reviewer oE5t). My intent with Section 2 was to explain how using a mixture of uniform distributions that *partition* the domain of each quaternion component (i.e., the bounds of the different uniform distributions are *fixed*) allows AQuaMaM to effectively model densities on $\text{\textbf{SO}}(3)$. To summarize my reasoning:
>
> 1. The distribution is expressive.
> 2. The distribution makes it easy to enforce the geometric constraints of the problem.
> 3. Lastly, the distribution is easy to optimize through the "language model loss", which is not the case for standard Mixture Density Networks (i.e., where the parameters of the distributions are output by the model). In standard Mixture Density Networks, the *raw* mixture proportions (i.e., not the log mixture proportions as in AQuaMaM) are used to calculate the loss, which introduces numerical challenges making standard Mixture Density Networks difficult to optimize (see some of the references in Makansi et al. (2019)).
>
> Can you please elaborate on why you don’t find this reasoning compelling? Particularly in light of the empirical results demonstrating the approach can effectively learn from data.
>
> >One would expect that problems like the ones examined would exhibit some amount of local smoothness that is not easily captured with the mixture of uniform construction. Another option could have been, for example to use a mixture of Gaussians for each conditional (as, for example, in RNADE MOG https://www.pure.ed.ac.uk/ws/portalfiles/portal/11874029/1068.pdf). Can the authors speak to this choice? Have they considered a mixture of Gaussians for the conditionals with appropriate rescaling (e.g. via softmax) to the constrained space?
>
> Because a mixture of Gaussians has a support from $-\infty$ to $\infty$, the distribution would generate samples that are not unit quaternions and is therefore not appropriate here. Can you please expand on your suggestion to rescale by a softmax? It’s not clear to me how that would work or what the resulting density would be.
>
> As for other mixture distributions, arguably the most "natural" choice for rotations is a mixture of Binghams, which was employed by Gilitschenski et al. (2019) and Deng et al. (2020)—two of the baselines that IPDF outperformed by a wide margin in Murphy et al. (2021).
>
> >The authors describe their method and characterize it relative to the work of Murphy 2021. Why do the authors choose this as the base of comparison? I was not previously familiar with this approach so was surprised to see it treated as a gold-standard baseline to beat. Without a compelling explanation for the choice of this baseline the empirical results do not stand on their own as impressive.
>
> To quote Reviewer KBHk, IPDF is a "strong SoTA baseline". IPDF was described in an ICML 2021 paper by researchers from Google, [was Tweeted by the highly popular AK Twitter account](https://twitter.com/_akhaliq/status/1403223498819047427), and was extended to relative pose prediction in [an ECCV 2022 paper](https://jasonyzhang.com/relpose/) by researchers from Carnegie Mellon, so the model has received considerable exposure in the pose estimation community. Additionally, several concurrent submissions to ICLR also use IPDF as a baseline, e.g., [[1](https://openreview.net/forum?id=pvrkJUkmto), [2](https://openreview.net/forum?id=jHA-yCyBGb)].

---

> > ### Author Response · Authors · 2022-11-10
> > **Response to Reviewer WNVz (continued)**
> >
> > >The empirical component of the validation is very limited. The authors could strength the paper with comparisons to additional methods.
> >
> > I mentioned in both the second paragraph of Section 1 and the first paragraph of Section 4 that, on a pose distribution estimation task, IPDF outperformed several baselines—including the model proposed in Gilitschenski et al. (2019), an ICLR 2019 paper—by a wide margin, which is why I only compared AQuaMaM to IPDF. I summarized these baselines in Section 1 to emphasize the fact that these approaches represent a range of modeling designs, and IPDF comfortably outperformed all of them.
> >
> > >There has been quite a lot of recent work in this space. Can the authors comment on advantages and disadvantages to previous work including:
> >
> > Diffusion/score-based models are an interesting approach, but, like IPDF, they require many passes through the network when sampling and calculating likelihoods (additionally, the likelihoods calculated by diffusion models are actually evidence lower bounds, unlike the likelihoods calculated by AQuaMaM, which are exact). Given that one of the primary strengths of diffusion models is their ability to model distributions for *high*-dimensional data (like images) where other approaches (like autoregressive models) struggle, it's not clear what, if any, advantages diffusion models would offer over AQuaMaM in the low-dimensional setting of pose estimation. Further, while AQuaMaM can be used to learn unconditional distributions, I was specifically interested in conditional distribution estimation (i.e., conditioning on an image). While conditional diffusion models exist, the diffusion/score-based papers you shared do not describe conditional models.
> >
> > The multimodal model described in Falorsi et al. (2019) is a normalizing flow. Normalizing flows may have limited expressivity due to the invertible transformation and easily computable Jacobian requirements (Kong and Chaudhuri, 2020). In contrast, Transformers are universal sequence approximators (Yun et al., 2020).
> >
> > Kong, Z., & Chaudhuri, K. (2020, June). The expressive power of a class of normalizing flow models. In International Conference on Artificial Intelligence and Statistics (pp. 3599-3609). PMLR.
> >
> > Yun, C., Bhojanapalli, S., Rawat, A. S., Reddi, S., & Kumar, S. (2020). Are Transformers universal approximators of sequence-to-sequence functions? International Conference on Learning Representations.
> >
> > >A chief advantage of the approach is said to be the computational speed-up relative to Murphy 2021. Is computation cost a limiting factor in the applications to which this method is applicable?
> >
> > Yes, the specific application I'm concerned with is estimating the poses of many individual weeds in a crop field as a tractor is driving over it, so throughput is important. Regardless, given any particular throughput tolerance (which will always exist; a pose estimation model that takes years to make a single prediction is not useful), AQuaMaM can be made larger/more expressive than its IPDF counterpart, which is advantageous when working with large datasets. Additionally, I want to emphasize that a second (equally, if not more important) advantage of AQuaMaM over IPDF is that the upper bound on the likelihood for AQuaMaM grows at least cubically with the number of output bins. In contrast, the upper bound for IPDF only grows linearly with the number of rotation grid cells (this advantage was mentioned in Section 2.4).

---

> > > ### Comment · Reviewer_WNVz · 2022-11-12
> > > **Additional responses.**
> > >
> > > Re-comparisons to other methods.  In figure 3 of this paper (https://openreview.net/pdf?id=jHA-yCyBGb) which you shared, the other baselines look better by eye than Murphy 2021.  Is there a reason why you have not compared to conditional variants of these models as baselines (i.e. RGSM, Leach and Moser flow)?
> > >
> > >
> > > Re-computation: Thank you for this clarification.  I believe that providing more explicit attention to the computational difficulty of Murphy 2021 as you have described would strengthen the paper.

---

> > ### Comment · Reviewer_WNVz · 2022-11-12
> > **response to rebuttal**
> >
> > Thank you for your thoughtful reply.
> >
> > Re- figure 7A.  Thank you for this clarification.  I had misunderstood this visualization and see now that the results are more compelling than I had originally thought. Do you have a sense of why the the likelihood of the two possibilities labeled `4` and `14` are so different (with a likelihood ratio over 100,000, if I understand correctly)?  Perhaps one clearer why to visualize this imbalance of between the possible solutions, and the clustering away from the other (even less realistic) parts of the space would be to additionally plot random samples from the distribution on SO(3).  I recommend adding this in.
> >
> > re-mixture of Gaussians: author authors (e.g. in the work I cited) have found mixtures of Gaussians easy to optimize.  I have also found them easier to optimize, and more accurate than mixtures of uniforms in my own work (not on SO(3)).
> >
> > I don't follow the authors reply about log proportions.  In MDNs one typically parameterizes log proportions and uses the softmax.
> >
> > By appropriate rescaling I meant that if a ~ MDN(x) then to constrain within [alpha, gamma] one can define say b = alpha + (gamma-alpha)*(e^a/(1+e^a)).  The log likelihood is then still available through a change of variables.
> >
> > Thank you for the additional background on IPDF.  Including some of this (e.g. in a brief appendix section) would have aided me as a reader.  Though I don't mind either way if you make such an addition.  The visual in that tweet is extremely neat, thank you for sharing.

---

> > > ### Author Response · Authors · 2022-11-13
> > > **Response #2 to Reviewer WNVz**
> > >
> > > Thank you for engaging in a dialogue and raising your score!
> > >
> > > >Do you have a sense of why the the likelihood of the two possibilities labeled 4 and 14 are so different (with a likelihood ratio over 100,000, if I understand correctly)?
> > >
> > > This appears to have been an artifact of my beam search. Other points close to point 14 have high log-likelihoods (the highest I saw was 12.15). To fix this ambiguity, I’ve updated Figure 7 so that the “positive” points are now sampled from the AQuaMaM distribution rather than obtained from a beam search.
> > >
> > > >Perhaps one clearer why to visualize this imbalance of between the possible solutions, and the clustering away from the other (even less realistic) parts of the space would be to additionally plot random samples from the distribution on SO(3). I recommend adding this in.
> > >
> > > I personally feel sampling can give a misleading impression of a model's distribution (particularly with regards to the relative frequencies of different modes), which is why I was interested in visualizing the model's density. However, at your request, I've added a plot (Figure 8 in the appendix) of 1,000 sampled points using the density shown in Figure 7a. As expected given the density plot, each of the equivalent modes is represented, and there are no points in incorrect regions.
> > >
> > > >re-mixture of Gaussians: author authors (e.g. in the work I cited) have found mixtures of Gaussians easy to optimize.
> > >
> > > I’m a little confused about what you’re saying here because the works you cited weren't using mixtures of Gaussians, or are you suggesting the referenced authors have used mixtures of Gaussians in other work?
> > >
> > > >I have also found them easier to optimize, and more accurate than mixtures of uniforms in my own work (not on SO(3)).
> > >
> > > Of course few things in machine learning are universal, and I'm not religiously opposed to Mixture Density Networks (MDNs), but the optimization challenges of MDNs have been widely documented, and MDNs are often outperformed by other models (which I suspect explains their relative rarity). Further, generating a single prediction with MDNs is more involved since (potentially many) samples need to be generated and evaluated.
> > >
> > > Additionally, I want to again emphasize that the mixture of uniforms being optimized by AQuaMaM is *not* an MDN, i.e., the model is *not* outputting the bounds of many (potentially overlapping) uniform distributions. The bounds of the uniform distributions in AQuaMaM are fixed and partition the domain of the relevant quaternion component. If AQuaMaM was an MDN, the model wouldn’t be able to use the language model loss trick I described. Lastly, I feel the empirical results are quite dispositive with regards to AQuaMaM's ability to learn $\text{\textbf{SO}}(3)$ distributions from data.
> > >
> > > >I don't follow the authors reply about log proportions. In MDNs one typically parameterizes log proportions and uses the softmax.
> > >
> > > The likelihood for a MDN is (see Equation (29) in Bishop (1994)):
> > >
> > > $p(x) = \sum_{i=1}^{M} \pi_{i} p_{i}(x)$
> > >
> > > where the $\pi_{i}$ terms are the mixture proportions and are indeed calculated with a softmax, while the $p_{i}$ terms are the densities for the mixture components. Taking the log of this likelihood gives you:
> > >
> > > $\ln{p(x)} = \ln{\sum_{i=1}^{M} \pi_{i} p_{i}(x)}$
> > >
> > > i.e., the *raw* mixture proportions are used in the log-likelihood, which can lead to the numerical instabilities I previously described. While there are various ways of addressing some of the numerical issues with MDNs (e.g., using the LogSumExp trick), NaNs and mode collapse are still frequently encountered problems.
> > >
> > > In contrast, the log-likelihood for AQuaMaM is:
> > >
> > > $\ln{p(x)} = \ln{\sum_{i=1}^{M} \pi_{i} p_{i}(x)} = \ln{\pi_{k} p_{k}(x)} =  \ln{\pi_{k}} + \ln{p_{k}(x)}$
> > >
> > > i.e., all of the terms except for the term corresponding to the bin containing $x$ drop out because the uniform distributions *partition* $x$’s domain. As a result, AQuaMaM is optimizing the *log* of the mixture proportion (just like a classifier), which is numerically stable.
> > >
> > > >By appropriate rescaling I meant that if a ~ MDN(x) then to constrain within [alpha, gamma] one can define say b = alpha + (gamma-alpha)*(e^a/(1+e^a)). The log likelihood is then still available through a change of variables.
> > >
> > > Thank you for clarifying. Just to summarize in my own words, what you’re suggesting is that $a$ is distributed according to an MDN with density $p_{M}(a)$, and $b = \alpha + (\gamma - \alpha) \frac{e^{a}}{1+e^{a}}$, i.e., $a$ determines where $b$ falls in $(\alpha, \gamma)$ through the logistic function. The density for $b$ can be calculated by a change of variables, so the final model in this case would consist of two density transformations: the first constraining the MDN output to the 3-ball, and the second being the same as in AQuaMaM. The only issue I see here is that the model is an MDN and will thus potentially face the previously discussed issues.

---

> > > > ### Author Response · Authors · 2022-11-13
> > > > **Response #2 to Reviewer WNVz (continued)**
> > > >
> > > > >Re-comparisons to other methods. In figure 3 of this paper (https://openreview.net/pdf?id=jHA-yCyBGb) which you shared, the other baselines look better by eye than Murphy 2021.
> > > >
> > > > I’m hesitant to draw any conclusions based on a task that IPDF wasn’t designed for and when using hyperparameters that were selected for a different task.
> > > >
> > > > >Is there a reason why you have not compared to conditional variants of these models as baselines (i.e. RGSM, Leach and Moser flow)?
> > > >
> > > > I previously mentioned that De Bortoli et al. (2022) and Leach et al. (2022) did not describe conditional models and were not performing pose estimation. The same is also true for Rozen et al. (2021). Extending these models to the conditional case for pose estimation would be a research project in itself. I hope the authors of these works will compare their models to AQuaMaM in the pose estimation setting in the future
> > > >
> > > > >Re-computation: Thank you for this clarification. I believe that providing more explicit attention to the computational difficulty of Murphy 2021 as you have described would strengthen the paper.
> > > >
> > > > I personally feel I was pretty explicit about the computational drawbacks of IPDF. From my abstract (which was reiterated in Section 1):
> > > >
> > > > >However, inference with IPDF requires $N$ forward passes through the network’s final multilayer perceptron—where $N$ places an upper bound on the likelihood that can be calculated by the model—which is prohibitively slow for those without the computational resources necessary to parallelize the queries.
> > > >
> > > > And from my summary at the end of Section 1:
> > > >
> > > > >AQuaMaM is *fast*, reaching a prediction throughput 52$\times$ faster than IPDF on a single GPU.
> > > >
> > > > Given the space limits for the paper, I think my comments on IPDF will have to serve as they are.

---

> > > > ### Comment · Reviewer_WNVz · 2022-11-13
> > > > **additional follow-ups on mixture of Gaussians**
> > > >
> > > > Re: "the works you cited weren't using mixtures of Gaussians".  Apologies for not being clear which works to which I was referring.  I meant this paper: https://proceedings.neurips.cc/paper/2013/hash/53adaf494dc89ef7196d73636eb2451b-Abstract.html, which I referred to as RNADE-MOG in my initial paper.  See also this follow-up work (hhttps://proceedings.neurips.cc/paper/2017/hash/6c1da886822c67822bcf3679d04369fa-Abstract.html) which is also quite widely cited / known to be effective.  I strongly encourage the authors to explore this form for conditionals as well (e.g. in further work), as I would be surprised if they encounter challenging optimization difficulties in their setting (which certainly are not experienced universally).
> > > >
> > > > Thank you for your clarifying summary of the proposal above.  This is indeed what I was intending to describe.
> > > >
> > > > As for the instability of the exponentiated probabilities, the standard tricks (e.g. log-sum-exp) may indeed be necessary to prevent underflow (if underflow if what you refer to when you say instability).  Or if you don't mean underflow, I don't understand the nature of the instability you describe as (again) I have not encountered it.
> > > >
> > > > Thank you for your updates to figures.  I find figure 8 compelling!

---

> > > > > ### Author Response · Authors · 2022-11-15
> > > > > **Mixture of Gaussians Experiment**
> > > > >
> > > > > I went ahead and implemented a mixture of Gaussians (MoG) AQuaMaM based on your proposed approach (full experiment details are described in Section A.9.2). I verified my MoG implementation was correct by optimizing just the MoG parameters on a toy dataset generated by four equally likely Gaussians with different means and variances. I’ve included the code for this verification experiment in the Supplementary Materials ZIP. The functions I used to: (1) sample from AQuaMaM-MoG, (2) calculate the AQuaMaM-MoG training loss, and (3) calculate the full AQuaMaM-MoG log-likelihood are also included in the code (`sample_aquamam_qcs`, `get_aquamam_mog_training_lls`, and `get_aquamam_mog_full_lls`). I trained an AQuaMaM-MoG model with 512 mixture components on the toy dataset described in Section 4.1 with otherwise identical hyperparameters to AQuaMaM. The LL on the test set for AQuaMaM-MoG was 10.52, which is not only far worse than AQuaMaM (27.12), but is also considerably worse than IPDF (12.32). I feel this evidence further justifies AQuaMaM's design.

---

> > > > > > ### Comment · Reviewer_WNVz · 2022-11-19
> > > > > > **Receipt of reply**
> > > > > >
> > > > > > Thank you for this reply.  I believe your changes have strengthened the paper as compared to the initial submission.

---

### Official Review · Reviewer_oE5t · 2022-10-26

**Confidence:** 3
**Correctness:** 3
**Technical Novelty And Significance:** 2
**Empirical Novelty And Significance:** 2
**Recommendation:** 3

**Clarity, Quality, Novelty And Reproducibility:**

The originality and novelty of the approach is questionable: This appears more like a standard model design choice for the particular application (you know that the phenomenon has rotational symmetry, and thus you incorporate this as prior knowledge in your model), and not as a new 'method'. The presentation could be improved. The text is wordy and could be stating things more directly and clearly. The paper has been written in first person ("I introduce...") which is non-standard and sounds a bit weird (this is a matter of taste, of course).

The author(s) have not shared code for replicating their experiments, but they provide pseudo-code in the appendix.


**Strength And Weaknesses:**

*Strengths*

1. Quaternions are the standard way of modelling rotations in tracking, position estimation, and pose modelling, and thus most likely a sensible way of modelling them also for capturing complex rotational distributions in general.

2. The paper is interesting, and could topic-wise be a good fit for the conference.

*Weaknesses*

3. The originality and novelty of the approach is questionable: This appears more like a standard model design choice for the particular application (you know that the phenomenon has rotational symmetry, and thus you incorporate this as prior knowledge in your model), and not as a new 'method'.

4. The presentation could be improved. The text is wordy and could be stating things more directly and clearly. The abstract alone almost fills the first page.

5. The experiments act as proof of concept. Additional experiments would have strengthened the paper.


**Summary Of The Paper:**

This paper uses a quaternion model for capturing rotational distributions. The approach is branded as a new 'method' even if it appears more like a model construction. The efficiency over a baseline is demonstrated on simple data sets.


**Summary Of The Review:**

This paper is interesting but appears to have multiple flaws.

---

> ### Author Response · Authors · 2022-11-10
> **Response to Reviewer oE5t**
>
> Thank you for your review. I’m glad we agree that the topic of the paper is a good fit for ICLR.
>
> >The author(s) have not shared code for replicating their experiments, but they provide pseudo-code in the appendix.
>
> I want to emphasize that the link to the repository mentioned in footnote 2 will be de-anonymized after the review process when the license can be included with the code.
>
> >The originality and novelty of the approach is questionable: This appears more like a standard model design choice for the particular application (you know that the phenomenon has rotational symmetry, and thus you incorporate this as prior knowledge in your model), and not as a new 'method'.
>
> Can you please elaborate on what you mean here? I only use the word "symmetry" once in the main text (when referring to the fact that some objects have continuous symmetry), and AQuaMaM does not use information about rotational symmetry anywhere in the architecture—it’s directly modeling a distribution on a manifold; the elements of the manifold just happen to correspond to rotations in this case. The novelty of AQuaMaM stems from its autoregressive mixture of uniforms design, which allows it to learn to explicitly model arbitrarily complex densities on $\text{\textbf{SO}}(3)$—a non-trivial task as demonstrated by prior work—simply by being trained as a “quaternion language model”. Given AQuaMaM’s significant improvement over the current state-of-the-art, I'm inclined to believe the architecture is not a “standard model design” since it was not previously described.
>
> >The presentation could be improved. The text is wordy and could be stating things more directly and clearly.
>
> I’ve made significant changes to the theory section to improve the presentation of the methods and intuition behind the approach. I hope this addresses your concerns. If not, because the other two reviewers both commented favorably on the quality of the writing, I would be grateful if you could provide specific examples of where the presentation should be improved.
>
> >The experiments act as proof of concept. Additional experiments would have strengthened the paper.
>
> Based on specific feedback from Reviewer KBHk, I’ve replicated the main die experiment using a cylinder (one of the objects from the SYMSOL I dataset) instead. For this dataset, the IPDF model reached an average log-likelihood (LL) of 5.94 on the test set while AQuaMaM reached an average LL of 7.24, a 21.7% improvement. I’ve replaced Section A.7 (now Section A.9.1) with a summary of this new experiment and a figure similar to Figure 7.
>
> Additionally, based on specific feedback from Reviewer WNVz, I've added an experiment training a mixture of Gaussians variant of AQuaMaM (described in Section A.9.2) on the toy dataset. The LL on the test set for AQuaMaM-MoG was 10.52, which is not only far worse than AQuaMaM (27.12), but is also considerably worse than IPDF (12.32).

---

### Official Review · Reviewer_KBHk · 2022-11-03

**Confidence:** 4
**Correctness:** 3
**Technical Novelty And Significance:** 3
**Empirical Novelty And Significance:** 3
**Recommendation:** 6

**Clarity, Quality, Novelty And Reproducibility:**

Clarity
-----
- Writing is in general very good. Section 1 is particularly well-written
- Some statements and technical steps were not clear or simple to understand:
   1) "IPDF is trained with Ntrain ≪ Ntest which can make it difficult to reason about how the model will behave in the wild" --> I don't see how the number of employed test points and its relationship with the number of training points could impact performance expectations on unseen data.
   2) In Sec. 2.2, I found it hard to understand the role of the unit 3-ball. While reading, it was non-trivial to grasp why "Therefore, there is a one-to-one mapping f : B3 → H1". There seemed to be a gap w.r.t. the previous paragarph: it was not immediately obvious how the remaining quaternion parameters $q_x, q_y, q_z$ map to $B^3$
   3) In Eq. (1), the reasoning leading from the second expression (with the summation) and the third one is not clear. It goes from a summation of weighted uniform distributions to a single term weighted by $\pi_k$.
   4) After Eq. (2), the derivation of the conditional distribution term $p(q_z \| q_x, q_y)$ is not provided. It may possibly be trivial (or not), but in my view it should be at least reported explicitly in the Appendix.
   5) The origin of $q_w$ at the numerator in (4) is not clear. What is its relationship with $s_q$? I'd suggest to make the passages explicit.
- Some important points refer the reader to prior works. It would help a lot to provide a synthetic description of relevant tools in the main paper or possibly in the appendix to facilitate understanding. In particular:
   1) The partially causal attention mask
   2) The claim that "models using mixtures of uniform distributions that partition some domain can be optimized solely using a “language model loss”"
- Not clear whether inference requires a single pass as stated in the abstract or three passes (although optimized via caching) as mentioned at the end of Section 3.2
- In general, figures help a lot in grasping some of the most complex passages. Still, possible improvements may include:
   1) Extending the scheme in Fig. 4 or adding a new one to illustrate the steps outlined in the last paragraph of Sec. 3.1, namely the generation of the transformed tokens and the classifier head;
   2) The description below the underbrace in Fig. 4 may possibly be wrong. The first tokens should be patch embeddings, while the last 3 should be the position embeddings as far as I understood;
   3) In Fig. 6b, the Category 4 subfigure should probably have 16 rotations, while only 3 are shown.
- There is some notational confusion and redundancy between $\pi_k, \pi_{q_c}, c, k$ and between $\omega_c$ and $\hat b_i$ from Figure 2
- Define $\pi_{q_{d,c}}$ and explain the difference from $\pi_{q_{c}}$
- Consider making the model parameters explicit in the NLL loss definition, i.e., $\mathcal{L}(\pi_{q_{d,c}}, \mathcal{X})$

Quality
----
- The quality of the work is generally high. The proposed architecture seems to be suitable for this kind of problem and is able to overcome some limitations of IPDF on well-designed experiments.

- Only one baseline is compared against (IPDF), but this choice appears justified by its strength with respect to other candidates (although on a different dataset).

- Training time to convergence is significantly faster for the proposed method w.r.t. IPDF, as shown in Fig.  5, supporting one of the main claims of this work

-  I agree on the fact that the proposed die experimental scenario is more controllable and allows to better study some key aspects of the method and the baseline, as stated by the author in Sec. 4. Still, an additional comparison on the symmetric solids dataset employed in the original IPDF paper may be informative to evaluate the comparative performance of AQuaMaM on a more challenging task.

Novelty
---
- The approach appears indeed innovative. I am not aware of methods employing Transformer architectures to tackle this problem

Reproducibility
----
The approach seems to be described in sufficient detail for possibly reimplementing it. Implementation details and code snippets are reported in the appendix.

**Strength And Weaknesses:**

Strengths
---------
- To my knowledge, the proposed approach is novel. It seems to be the first successful application of a Transformer autoregressive model for multimodal distribution estimation on SO(3).
- AQuaMaM is compared against a strong baseline (IPDF), demonstrating:
   1) Significantly faster convergence time
   2) Faster inference, since it requires few forward passes with respect to the $N$ required by IPDF
   3) Higher accuracy
   4) Higher reliability (see Fig. 6)
- The paper is generally well-written. The introduction is particularly pleasant to read and positions the paper very well in the literature.

Weaknesses
------------
- As detailed in the following, there are several points in which clarity shall be improved. This would especially help readers who may be familiar with the orientation estimation problem, yet not necessarily accustomed to language modeling and Transformers.
- The die experiment is convincing and allows for detailed analysis in a controlled way. Still, an additional comparison on a more challenging dataset (e.g., the one employed in the original IPDF paper) would strengthen the paper by validating AQuaMaM on an additional, possibly harder problem.

**Summary Of The Paper:**

This paper presents a novel method (AQuaMaM) for 3D quaternion orientation estimation from potentially ambiguous 2D images. It employs a Transformer architecture to learn sequences of quaternion parameters representing distributions over the SO(3) group of 3D rotations, treating them as language tokens.

The proposed architecture is able to efficiently learn multimodal distributions over SO(3), allowing to represent multiple legitimate candidate rotations corresponding to a given ambiguous 2D image.

The approach is validated on a toy dataset and a 3D die orientation estimation dataset, demonstrating higher accuracy and higher training and inference time efficiency with respect to a strong SoTA baseline (IPDF).

**Summary Of The Review:**

Overall, in my view the paper makes a significant algorithmic contribution by presenting a novel Transformer-based approach to complex multi-modal distribution learning on SO(3). The proposed approach is demonstrated to be faster and more accurate and reliable than a strong baseline on two clear and controllable experimental setups.

The paper is generally well-written and pleasant to read. Some improvements would be required in terms of clarity, as detailed above.

---

> ### Author Response · Authors · 2022-11-10
> **Response to Reviewer KBHk**
>
> Thank you for your thoughtful and detailed review! I sincerely appreciate the effort you put into providing constructive feedback. I’ve attempted to incorporate as much of your feedback as possible, and I believe the manuscript has improved because of it.
>
> >The approach seems to be described in sufficient detail for possibly reimplementing it. Implementation details and code snippets are reported in the appendix.
>
> I want to emphasize that the link to the repository mentioned in footnote 2 will be de-anonymized after the review process when the license can be included with the code. I think the code will serve as a nice pedagogical complement to the Transformer background I’ve added to the manuscript (described below).
>
> >This would especially help readers who may be familiar with the orientation estimation problem, yet not necessarily accustomed to language modeling and Transformers.
>
> I admittedly assumed some familiarity with Transformers due to my experience with them and their ubiquity in other fields. At your request, I’ve added a (two page) section (Section A.3) to the appendix that links to several excellent pedagogical materials on Transformers (namely, [The Illustrated Transformer](https://jalammar.github.io/illustrated-transformer/), [The Annotated Transformer](http://nlp.seas.harvard.edu/annotated-transformer/), and [Attention? Attention!](https://lilianweng.github.io/posts/2018-06-24-attention/)), and briefly introduces the Transformer architecture, being sure to define the Transformer-specific terms used in the main text (specifically, “position embeddings” and “partially causal attention mask”). I feel these additions will allow individuals who are new to Transformers to quickly acquire the necessary background to fully understand the AQuaMaM architecture.
>
> >The description below the underbrace in Fig. 4 may possibly be wrong. The first tokens should be patch embeddings, while the last 3 should be the position embeddings as far as I understood;
>
> As noted above, “position embeddings” is a Transformer-specific term and does not refer to the quaternion component embeddings. Position embeddings are used to indicate where in a sequence a particular element is found. The new Transformer section gives an explicit example of how position embeddings are used. However, I’ve removed the “+ Position Embeddings” text on the figure to eliminate any chance for confusion, and because position embeddings are closely tied to the Transformer algorithm, so they can reasonably be assumed.
>
> >Not clear whether inference requires a single pass as stated in the abstract or three passes (although optimized via caching) as mentioned at the end of Section 3.2
>
> Calculating the likelihood for a specific quaternion only requires a single forward pass through AQuaMaM, but making a prediction requires three forward passes. The new Transformer section makes this distinction clear.
>
> >Extending the scheme in Fig. 4 or adding a new one to illustrate the steps outlined in the last paragraph of Sec. 3.1, namely the generation of the transformed tokens and the classifier head;
>
> I think this procedure is actually easiest to understand through pseudocode, so I've added PyTorch-like pseudocode to Section A.6 to help make these steps more clear.
>
> >In Sec. 2.2, I found it hard to understand the role of the unit 3-ball. While reading, it was non-trivial to grasp why "Therefore, there is a one-to-one mapping f : B3 → H1". There seemed to be a gap w.r.t. the previous paragarph: it was not immediately obvious how the remaining quaternion parameters $q_{x}$, $q_{y}$, $q_{z}$  map to $B^{3}$
>
> I’ve significantly restructured Section 2 to help clarify the manifold structure of the "hyper-hemisphere" and how it relates to the 3-ball. Specifically, I slightly reworked the first paragraph of the old Section 2.2, moved that paragraph into Section 2.3, moved Section 2.3 to Section 2.2, and tried to explain how all the pieces fit together a little differently.
>
> >The origin of $q_{w}$  at the numerator in (4) is not clear. What is its relationship with $s_{q}$? I'd suggest to make the passages explicit.
>
> While I did explicitly perform this calculation in Section A.2 for the hemisphere example depicted in Figure 3 (showing $a = \frac{1}{z}$ and so $\frac{1}{a} = z$), you're right that I never explicitly stated how $s_{q}$ is calculated, only saying that the calculation was "directly analogous" to the hemisphere example. I've added a sentence explicitly stating its value ($\frac{1}{q_{w}}$) to Section 2.2, and I’ve added two sentences to Section A.1 explaining the small way its calculation differs from the hemisphere example.

---

> > ### Author Response · Authors · 2022-11-10
> > **Response to Reviewer KBHk (continued)**
> >
> > >In Eq. (1), the reasoning leading from the second expression (with the summation) and the third one is not clear. It goes from a summation of weighted uniform distributions to a single term weighted by $\pi_{k}$.
> >
> > The bounds of the bins corresponding to the uniform distributions (indexed by $i$) [*partition*](https://en.wikipedia.org/wiki/Partition_of_a_set) $[-1, 1]$ (which is mentioned two sentences prior to Equation (1)). As a result, $q_{x}$ will only fall within the bounds of *one* of the uniform distributions. Because a uniform distribution assigns a density of zero to points that fall outside of its bounds (which is mentioned in the sentence prior to Equation (1)), all of the terms corresponding to bins not containing $q_{x}$ drop out.
> >
> > >After Eq. (2), the derivation of the conditional distribution term $p(q_{z}|q_{x},q_{y})$ is not provided. It may possibly be trivial (or not), but in my view it should be at least reported explicitly in the Appendix.
> >
> > $p(q_{z}|q_{x},q_{y})$ is indeed almost identical to Equation (2), but I’ve added its definition to be explicit.
> >
> > >The die experiment is convincing and allows for detailed analysis in a controlled way. Still, an additional comparison on a more challenging dataset (e.g., the one employed in the original IPDF paper) would strengthen the paper by validating AQuaMaM on an additional, possibly harder problem.
> >
> > >Still, an additional comparison on the symmetric solids dataset employed in the original IPDF paper may be informative to evaluate the comparative performance of AQuaMaM on a more challenging task.
> >
> > I did include a mini experiment in Section A.7 to demonstrate that AQuaMaM is capable of learning distributions corresponding to continuous symmetries; however, at your request, I’ve replicated the main die experiment using a cylinder (one of the objects from the SYMSOL I dataset) instead. My personal intuition is that the lack of distinguishing features on the SYMSOL I objects makes the task easier because there’s less variety in the input images (i.e., more poses produce exactly the same image compared to the die). Regardless, for this dataset, the IPDF model reached an average log-likelihood (LL) of 5.94 on the test set (compared to 4.26 in Murphy et al. (2021) where they used a smaller training dataset and updated the model fewer times during training) while AQuaMaM reached an average LL of 7.24, a 21.7% improvement. I’ve replaced Section A.7 (now Section A.9.1) with a summary of this new experiment and a figure similar to Figure 7.
> >
> > >"IPDF is trained with $N_{train} \ll N_{test}$ which can make it difficult to reason about how the model will behave in the wild" --> I don't see how the number of employed test points and its relationship with the number of training points could impact performance expectations on unseen data.
> >
> > $N$ is the number of pieces that $\text{\textbf{SO}}(3)$ has been split into, not the number of training samples (if I’m understanding your use of “training points” correctly). Because the training query rotations are not the same as those that are used to make a prediction, it’s difficult (at least for me) to reason about how the output distribution for a test image will behave. My intent with the toy dataset experiment and Figure 6 was to highlight this reasoning gap.
> >
> > >The claim that "models using mixtures of uniform distributions that partition some domain can be optimized solely using a “language model loss”
> >
> > The equivalence is described in the text following Equation (5), i.e., it’s due to the fact that the diluting factors are constant for a given dataset and can thus be ignored during optimization. Is there something specific you would like to see further explained?
> >
> > >There is some notational confusion and redundancy between $\pi_{k}$, $\pi_{q_{c}}$, $c$, $k$ and between $\omega_{c}$ and $\hat{b}_{i}$ from Figure 2
> >
> > I took another look at Figure 2 (now Figure 3) and I couldn’t see where the notation was confusing or redundant. Can you please be more specific? I’ve corrected $\omega_{c}$ to $\omega_{q_{c}}$ in Section 2.3.
> >
> > >In Fig. 6b, the Category 4 subfigure should probably have 16 rotations, while only 3 are shown.
> >
> > Yes, the remaining 13 rotations weren't shown because they didn't have any samples (likewise with Category 5 and the remaining 14 rotations). I’ve modified these figures to include the rotations with zero samples for clarity.
> >
> > >Define $\pi_{d,c}$  and explain the difference from $\pi_{c}$
> >
> > I’ve added a parenthetical explaining that $d$ is the index for a sample from the dataset.

---

> > > ### Comment · Reviewer_KBHk · 2022-11-17
> > > **Answer to the author**
> > >
> > > I would like to sincerely thank the Author for their detailed responses and improvements to the manuscript.
> > >
> > > At the moment, my evaluation remains "weak accept" (6), confirming the above-threshold rating due to my belief on the good overall quality and novelty of the work.
> > >
> > > I will consider and discuss the provided responses also during the next review phase and reserve the right to possibly update my evaluation.

---

> > > > ### Comment · Reviewer_KBHk · 2022-11-27
> > > > **Above-threshold Score Confidence Increased**
> > > >
> > > > I have read the conversations between the Author and the other Reviewers, together with the revised version of the paper.
> > > >
> > > > I believe the Author satisfactorily answered several relevant concerns and successfully explained unclear points. I also appreciate the additional experiments (especially the cylinder experiment, demonstrating the capabilities of the method in presence of continuous symmetries), although in general the empirical validation could have been more exhaustive in terms of baselines and datasets.
> > > >
> > > > Based on these considerations, I decided to increase the confidence in my above-threshold score from 3 to 4.

---

### Author Response · Authors · 2022-11-10
**Comment on AQuaMaM's Design**

I want to note that the characterizations of AQuaMaM's design by Reviewer oE5t and Reviewer WNVz appear to be incompatible. Reviewer oE5t seems to be suggesting the architecture is almost trivially obvious given the problem domain while Reviewer WNVz seems to feel the architecture is unmotivated. I believe if we can resolve these conflicting perspectives through a discussion, the reviewers will come to appreciate both the novelty and utility of AQuaMaM as was recognized by Reviewer KBHk.

---

### Author Response · Authors · 2023-02-02
**Code**

As promised, the code for the paper can be found [here](https://github.com/airalcorn2/aquamam).

---

### Author Response · Authors · 2023-03-12
**On Treating implicit-PDF as the "Gold Standard"**

I thought it was worth documenting that the paper "[Image to Sphere: Learning Equivariant Features for Efficient Pose Prediction](https://openreview.net/forum?id=_2bDpAtr7PI)" was not only accepted to ICLR 2023 but received a rating of “Accept: notable-top-5%” despite the fact it treats implicit-PDF as the "gold standard" for distribution estimation of poses (see Table 3). On top of that, implicit-PDF outperformed Image to Sphere by a wide margin!

---

### Decision · Program_Chairs · 2023-01-20

**Decision:**

Reject

**Justification For Why Not Higher Score:**

This is not a bad paper, but it could be much better if it did a proper empirical validation.

**Justification For Why Not Lower Score:**

n/a

**Metareview: Summary, Strengths And Weaknesses:**

After discussions, the reviewers have converged on ratings 6 (KBHk), 5 (WNVz), 3 (oE5t), which would generally put the paper below the threshold of acceptance. Reviewers KBHk and WNVz engaged in extensive discussions with the author, but unfortunately oE5t did not engage. Personally I found the paper to be fairly well-written, as noted also by some of the reviews. Furthermore, the method enjoys fast inference, convergence, and accuracy. Although many details were discussed, the most significant issues are related to methodological novelty / originality and experimental validation.

Originality: I agree with oE5t and WNVz that the idea of applying an autoregressive model to the (discretized) coordinates of a quaternion is not super original. However, simplicity is also a virtue, and it could well be that this approach has advantages over earlier more complex methods. This would however need to be convincingly demonstrated.

Experiments: the experiments are limited to synthetic data with limited variability (e.g. just a rotated dice). I agree with the reviewers who noted that it would be nice to see experiments involving more challenging / real data. The vision, robotics, and autonomous driving communities have worked on pose estimation for a long time, and have various benchmark datasets and methods. Furthermore, to convincingly demonstrate the benefits of the presented method, a comparison to a wider variety of baselines would be preferred.

The paper only compares to IPDF, and as WNVz notes it is not clear that this should be considered the gold standard. The authors argue that they outperform IPDF and IPDF outperforms various baselines, so that direct comparisons to other baselines is not required. I do not agree with this line of reasoning though, because IPDF outperforms the baselines on a dataset they introduced (also synthetic), and the present paper outperforms IPDF on a newly introduced dataset (rotated dice), so transitive reasoning does not apply. A truly convincing demonstration of the performance of AQuaMaM would involve comparing a number of competitive and popular baseline methods on a number of real-world datasets.

My conclusion is that this is quite a promising work, and I do not agree that the paper is a "clear reject". However, the paper could be made much more convincing by systematically evaluating on a range of datasets, and directly comparing to a range of baseline methods evaluated under identical conditions. As such I encourage the author to continue to improve the work and to submit it to the next conference.

**Summary Of Ac-Reviewer Meeting:**

There was no AC-reviewer meeting, as based on review scores this paper didn't make the cutoff to be a borderline paper. However based on my own detailed assessment of the paper at a later point, I do consider it borderline, as discussed in my meta review.